# Smoothed Analysis of Online and Differentially Private Learning

**Nika Haghtalab**
Department of Computer Science
Cornell University
nika@cs.cornell.edu

**Tim Roughgarden**
Department of Computer Science
Columbia University
tr@cs.columbia.edu

**Abhishek Shetty**
Department of Computer Science
Cornell University
shetty@cs.cornell.edu

## Abstract

Practical and pervasive needs for robustness and privacy in algorithms have inspired the design of online adversarial and differentially private learning algorithms. The primary quantity that characterizes learnability in these settings is the Littlestone dimension of the class of hypotheses [Alon et al., 2019, Ben-David et al., 2009]. This characterization is often interpreted as an impossibility result because classes such as linear thresholds and neural networks have infinite Littlestone dimension. In this paper, we apply the framework of smoothed analysis [Spielman and Teng, 2004], in which adversarially chosen inputs are perturbed slightly by nature. We show that fundamentally stronger regret and error guarantees are possible with smoothed adversaries than with worst-case adversaries. In particular, we obtain regret and privacy error bounds that depend only on the VC dimension and the bracketing number of a hypothesis class, and on the magnitudes of the perturbations.

## 1 Introduction

Robustness to changes in the data and protecting the privacy of data are two of the main challenges faced by machine learning and have led to the design of *online* and *differentially private* learning algorithms. While offline PAC learnability is characterized by the finiteness of VC dimension, online and differentially private learnability are both characterized by the finiteness of the Littlestone dimension [Alon et al., 2019, Ben-David et al., 2009, Bun et al., 2020]. This latter characterization is often interpreted as an impossibility result for achieving robustness and privacy on worst-case instances, especially in classification where even simple hypothesis classes such as 1-dimensional thresholds have constant VC dimension but infinite Littlestone dimension.

Impossibility results for worst-case adversaries do not invalidate the original goals of robust and private learning with respect to practically relevant hypothesis classes; rather, they indicate that a new model is required to provide rigorous guidance on the design of online and differentially private learning algorithms. In this work, we go beyond worst-case analysis and *design online learning algorithms and differentially private learning algorithms as good as their offline and non-private PAC learning counterparts in a realistic semi-random model of data.*

Inspired by smoothed analysis [Spielman and Teng, 2004], we introduce frameworks for online and differentially private learning in which adversarially chosen inputs are perturbed slightly by nature (reflecting, e.g., measurement errors or uncertainty). Equivalently, we consider an adversary restricted to choose an input distribution that is not overly concentrated, with the realized input then drawn

from the adversary's chosen distribution. Our goal is to design algorithms with good expected regret and error bounds, where the expectation is over nature's perturbations (and any random coin flips of the algorithm). Our positive results show, in a precise sense, that the known lower bounds for worst-case online and differentially private learnability are fundamentally brittle.

**Our Model.** Let us first consider the standard online learning setup with an instance space $\mathcal{X}$ and a set $\mathcal{H}$ of binary hypotheses each mapping $\mathcal{X}$ to $\mathcal{Y} = \{+1, -1\}$. Online learning is played over $T$ time steps, where at each step the learner picks a prediction function from a distribution and the *adaptive* adversary chooses a pair of $(x_t, y_t) \in \mathcal{X} \times \mathcal{Y}$. The regret of an algorithm is the difference between the number of mistakes the algorithm makes and that of the best fixed hypothesis in $\mathcal{H}$. The basic goal in online learning is to obtain a regret of $o(T)$. In comparison, in differential privacy the data set $B = \{(x_1, y_1), \ldots, (x_n, y_n)\}$ is specified ahead of time. Our goal here is to design a randomized mechanism that with high probability finds a nearly optimal hypothesis in $\mathcal{H}$ on the set $B$, while ensuring that the computation is *differentially private*. That is, changing a single element of $B$ does not significantly alter the probability with which our mechanism selects an outcome. Similar to agnostic PAC learning, this can be done by ensuring that the error of each hypothesis $h \in \mathcal{H}$ on $B$ (referred to as a query) is calculated accurately and privately.

We extend these two models to accommodate smoothed adversaries. We say that a distribution $\mathcal{D}$ over instance-label pairs is $\sigma$-*smooth* if its density function over the instance domain is pointwise bounded by at most $1/\sigma$ times that of the uniform distribution. In the online learning setting this means that at step $t$, the adversary chooses an arbitrary $\sigma$-smooth distribution $\mathcal{D}_t$ from which $(x_t, y_t) \sim \mathcal{D}_t$ is drawn. In the differential privacy setting, we work with a database $B$ for which the answers to the queries could have been produced by a $\sigma$-smooth distribution.

While we assume that $1/\sigma$ is slowly growing (subexponentiallly) in $T$, we can use values of $\sigma$ that depend on the dimension of the space to account for the volume of high dimensional domains. Since our bounds are only logarithmic in $1/\sigma$, they gracefully scale with the dimension of the space.

Note that this notion of bounded densities also incorporates smoothness models where worst-case instances are perturbed with small amount of noise, since convolution with noise produces distributions with bounded density. One advantage is that our smoothing model treats combinatorial domains (say $[n]$ or graphs on $n$ vertices) and geometric domains (say $D \subseteq \mathbb{R}^d$) in a unified manner and allows us to deliver results most meaningful to the analysis of learnability in presence of some smoothness without getting bogged down with domain-specific definitions of smoothness. Another advantage of our model — which is specifically important in machine learning — is that it naturally allows the adversary to have arbitrary correlations between the labels and the instances as long as the marginal on the instances is a "smooth" distribution. This can be handled by other models, albeit with more awkwardness in separating how an instance is generated by random shifts but its label is generated exactly by the adversary.

Why should smoothed analysis help in online learning? Consider the well-known lower bound for 1-dimensional thresholds over $\mathcal{X} = [0, 1]$, in which the learner may as well perform binary search and the adversary selects an instance within the uncertainty region of the learner that causes a mistake. While the learner's uncertainty region is halved each time step, the worst-case adversary can use ever-more precision to force the learner to make mistakes indefinitely. On the other hand, a $\sigma$-smoothed adversary effectively has bounded precision. That is, once the width of the uncertainty region drops below $\sigma$, a smoothed adversary can no longer guarantee that the chosen instance lands in this region. Similarly for differential privacy, there is a $\sigma$-smooth distribution that produces the same answers to the queries. Such a distribution has no more than $\alpha$ probability over an interval of width $\sigma\alpha$. So one can focus on computing the errors of the $1/(\sigma\alpha)$ hypotheses with discreized thresholds and learn a hypothesis of error at most $\alpha$. Analogous observations have been made in prior works (Rakhlin et al. [2011], Cohen-Addad and Kanade [2017], Gupta and Roughgarden [2017]), although only for very specific settings (online learning of 1-dimensional thresholds, 1-dimensional piecewise constant functions, and parameterized greedy heuristics for the maximum weight independent set problem, respectively). Our work is the first to demonstrate the breadth of the settings in which fundamentally stronger learnability guarantees are possible for smoothed adversaries than for worst-case adversaries.

**Our Results and Contributions.**

- Our main result concerns online learning with *adaptive $\sigma$-smooth* adversaries where $\mathcal{D}_t$ can depend on the history of the play, including the earlier realizations of $x_\tau \sim \mathcal{D}_\tau$ for $\tau < t$. That is, $x_t$ and $x_{t'}$ can be highly correlated. We show that regret against these powerful adversaries is bounded by $\tilde{O}(\sqrt{T \ln(\mathcal{N})})$, where $\mathcal{N}$ is the *bracketing number* of $\mathcal{H}$ with respect to the uniform distribution.[1] Bracketing number is the size of an $\epsilon$-cover of $\mathcal{H}$ with the additional property that hypotheses in the cover are *pointwise* approximations of those in $\mathcal{H}$. We show that for many hypothesis classes, the bracketing number is nicely bounded as a function of the VC dimension. This leads to the regret bound of $\tilde{O}(\sqrt{T \, \mathrm{VCDim}(\mathcal{H}) \ln(1/\sigma)})$ for commonly used hypothesis classes in machine learning, such as halfspaces, polynomial threshold functions, and polytopes. In comparison, these hypothesis classes have infinite Littlestone dimension and thus cannot be learned with regret $o(T)$ in the worst case [Ben-David et al., 2009].

  From a technical perspective, we introduce a novel approach for bounding time-correlated non-independent stochastic processes over infinite hypothesis classes using the notion of bracketing number. Furthermore, we introduce systematic approaches, such as high-dimensional linear embeddings and $k$-fold operations, for analyzing the bracketing number of complex hypothesis classes. We believe these techniques are of independent interest.

- For differentially private learning, we obtain an error bound of $\tilde{O}\big( \ln^{\frac{3}{8}}(1/\sigma) \sqrt{\mathrm{VCDim}(\mathcal{H})/n} \big)$; the key point is that this bound is independent of the size $|\mathcal{X}|$ of the domain and the size $|\mathcal{H}|$ of the hypothesis class. We obtain these bounds by modifying two commonly used mechanisms in differential privacy, the Multiplicative Weight Exponential Mechanism of Hardt et al. [2012] and the SmallDB algorithm of Blum et al. [2008]. With worst-case adversaries, these algorithms achieve only error bounds of $\tilde{O}(\ln^{\frac{1}{4}}(|\mathcal{X}|) \sqrt{\ln(|\mathcal{H}|)/n})$ and $\tilde{O}(\sqrt[3]{\mathrm{VCDim}(\mathcal{H}) \ln(|\mathcal{X}|)/n})$, respectively. Our results also improve over those in Hardt and Rothblum [2010] which concern a similar notion of smoothness and achieve an error bound of $\tilde{O}(\ln^{\frac{1}{2}}(1/\sigma) \sqrt{\ln(|\mathcal{H}|)/n})$.

**Other Related Works.** At a higher level, our work is related to several works on the intersection of machine learning and beyond the worst-case analysis of algorithms (e.g., [Balcan et al., 2018, Dekel et al., 2017, Kannan et al., 2018]) that are covered in more detail in Appendix A.

## 2 Preliminaries

**Online Learning.** We consider a measurable instance space $\mathcal{X}$ and the label set $\mathcal{Y} = \{+1, -1\}$. Let $\mathcal{H}$ be a hypothesis class on $\mathcal{X}$ with its VC dimension denoted by $\mathrm{VCDim}(\mathcal{H})$. Let $\mathcal{U}$ be the uniform distribution over $\mathcal{X}$ with density function $u(\cdot)$. For a distribution $\mathcal{D}$ over $\mathcal{X} \times \mathcal{Y}$, let $p(\cdot)$ be the *probability density function* of its marginal over $\mathcal{X}$. We say that $\mathcal{D}$ is $\sigma$-*smooth* if for all $x \in \mathcal{X}$, $p(x) \leq u(x)\sigma^{-1}$. For a labeled pair $s = (x, y)$ and a hypothesis $h \in \mathcal{H}$, $\mathrm{err}_s(h) = 1(h(x) \neq y)$ indicates whether $h$ makes a mistake on $s$.

We consider the setting of *online adversarial and (full-information) learning*. In this setting, a learner and an adversary play a repeated game over $T$ time steps. In every time step $t \in [T]$ the learner picks a hypothesis $h_t$ and adversary picks a $\sigma$-smoothed distribution $\mathcal{D}_t$ from which a labeled pair $s_t = (x_t, y_t)$ such that $s_t \sim \mathcal{D}_t$ is generated. The learner then incurs penalty of $\mathrm{err}_{s_t}(h_t)$. We consider two types of adversaries. First (and the subject of our main results) is called an *adaptive $\sigma$-smooth adversary*. This adversary at every time step $t \in [T]$ chooses $\mathcal{D}_t$ based on the actions of the learner $h_1, \ldots, h_{t-1}$ and, importantly, the realizations of the previous instances $s_1, \ldots, s_{t-1}$. We denote this adaptive random process by $\mathbf{s} \sim \mathscr{D}$. A second and less powerful type of adversary is called a *non-adaptive $\sigma$-smooth adversary*. Such an adversary first chooses an unknown sequence of distributions $\boldsymbol{\mathcal{D}} = (\mathcal{D}_1, \ldots, \mathcal{D}_T)$ such that $\mathcal{D}_t$ is a $\sigma$-smooth distribution for all $t \in [T]$. Importantly, $\mathcal{D}_t$ does not depend on realizations of adversary's earlier actions $s_1, \ldots, s_{t-1}$ or the learner's actions $h_1, \ldots, h_{t-1}$. We denote this non-adaptive random process by $\mathbf{s} \sim \boldsymbol{\mathcal{D}}$. With a slight abuse of notation, we denote by $\mathbf{x} \sim \mathscr{D}$ and $\mathbf{x} \sim \boldsymbol{\mathcal{D}}$ the sequence of (unlabeled) instances in $\mathbf{s} \sim \mathscr{D}$ and $\mathbf{s} \sim \boldsymbol{\mathcal{D}}$.

Our goal is to design an online algorithm $\mathcal{A}$ such that expected regret against an adaptive adversary,

$$\mathbb{E}[\text{REGRET}(\mathcal{A}, \mathscr{D})] := \mathbb{E}_{\mathbf{s} \sim \mathscr{D}} \left[ \sum_{t=1}^{T} \text{err}_{s_t}(h_t) - \min_{h \in \mathcal{H}} \sum_{t=1}^{T} \text{err}_{s_t}(h) \right]$$

is sublinear in $T$. We also consider the regret of an algorithm against a non-adaptive adversary defined similarly as above and denoted by $\mathbb{E}[\text{REGRET}(\mathcal{A}, \mathcal{D})]$.

**Differential Privacy.** We also consider differential privacy. In this setting, a data set $S$ is a multiset of elements from domain $\mathcal{X}$. Two data sets $S$ and $S'$ are said to be *adjacent* if they differ in at most one element. A randomized algorithm $\mathcal{M}$ that takes as input a data set is $(\epsilon, \delta)$-differentially private if for all $\mathcal{R} \subseteq \text{Range}(\mathcal{M})$ and for all adjacent data sets $S$ and $S'$, $\Pr[\mathcal{M}(S) \in \mathcal{R}] \leq \exp(\epsilon) \Pr[\mathcal{M}(S') \in \mathcal{R}] + \delta$. If $\delta = 0$, the algorithm is said to be purely $\epsilon$-differentially private.

For differentially private learning, one considers a fixed class of *queries* $\mathcal{Q}$. The learner's goal is to evaluate these queries on a given data set $S$. For ease of notation, we work with the empirical distribution $\mathcal{D}_S$ corresponding to a data set $S$. Then the learner's goal is to approximately compute $q(\mathcal{D}_S) = \mathbb{E}_{x \sim \mathcal{D}_S}[q(x)]$ while preserving privacy[2]. We consider two common paradigms of differential privacy. First, called *query answering*, involves designing a mechanism that outputs values $v_q$ for all $q \in \mathcal{Q}$ such that with probability $1 - \beta$ for every $q \in Q$, $|q(\mathcal{D}_S) - v_q| \leq \alpha$. The second paradigm, called *data release*, involves designing a mechanism that outputs a synthetic distribution $\overline{\mathcal{D}}$, such that with probability $1 - \beta$ for all $q \in \mathcal{Q}$, $|q(\overline{\mathcal{D}}) - q(\mathcal{D}_S)| \leq \alpha$. That is, the user can use $\overline{\mathcal{D}}$ to compute the value of any $q(\mathcal{D}_S)$ approximately.

Analogous to the definition of smoothness in online learning, we say that a distribution $\mathcal{D}$ with density function $p(\cdot)$ is $\sigma$-*smooth* if $p(x) \leq \sigma^{-1} u(x)$ for all $x \in \mathcal{X}$. We also work with a weaker notion of smoothness of data sets. A data set $S$ is said to be $(\sigma, \chi)$-*smooth* with respect to a query set $\mathcal{Q}$ if there is a $\sigma$-smooth distribution $\mathcal{D}$ such that for all $q \in \mathcal{Q}$, we have $|q(\mathcal{D}) - q(\mathcal{D}_S)| \leq \chi$. The definition of $(\sigma, \chi)$-smoothness, which is also referred to as *pseudo-smoothness* by Hardt and Rothblum [2010], captures data sets that though might be concentrated on some elements, the query class is not capable of noticing their lack of smoothness.

**Additional Definitions.** Let $\mathcal{H}$ be a hypothesis class and let $\mathcal{D}$ be a distribution. $\mathcal{H}'$ is an $\epsilon$-cover for $\mathcal{H}$ under $\mathcal{D}$ if for all $h \in \mathcal{H}$, there is a $h' \in \mathcal{H}'$ such that $\Pr_{x \sim \mathcal{D}}[h(x) \neq h'(x)] \leq \epsilon$. For any $\mathcal{H}$ and $\mathcal{D}$, there an $\epsilon$-cover $\mathcal{H}' \subseteq \mathcal{H}$ under $\mathcal{D}$ such that $|\mathcal{H}'| \leq (41/\epsilon)^{\text{VCDim}(\mathcal{H})}$ (Haussler [1995]).

We define a partial order $\preceq$ over functions such that $f_1 \preceq f_2$ if and only if for all $x \in \mathcal{X}$, we have $f_1(x) \leq f_2(x)$. For a pair of functions $f_1, f_2$ such that $f_1 \preceq f_2$, a *bracket* $[f_1, f_2]$ is defined by $[f_1, f_2] = \{f : \mathcal{X} \to \{-1, 1\} : f_1 \preceq f \preceq f_2\}$. Given a measure $\mu$ over $\mathcal{X}$, a bracket $[f_1, f_2]$ is called an $\epsilon$-bracket if $\Pr_{x \sim \mu}[f_1(x) \neq f_2(x)] \leq \epsilon$.

**Definition 2.1** (Bracketing Number). *Consider an instance space $\mathcal{X}$, measure $\mu$ over this space, and hypothesis class $\mathcal{F}$. A set $\mathcal{B}$ of brackets is called an $\epsilon$-bracketing of $\mathcal{F}$ with respect to measure $\mu$ if all brackets in $\mathcal{B}$ are $\epsilon$-brackets with respect to $\mu$ and for every $f \in \mathcal{F}$ there is $[f_1, f_2] \in \mathcal{B}$ such that $f \in [f_1, f_2]$. The $\epsilon$-bracketing number of $\mathcal{F}$ with respect to measure $\mu$, denoted by $\mathcal{N}_{[]}(\mathcal{F}, \mu, \epsilon)$, is the size of the smallest $\epsilon$-bracketing for $\mathcal{F}$ with respect to $\mu$.*

## 3 Regret Bounds for Smoothed Adaptive and Non-Adaptive Adversaries

In this section, we obtain regret bounds against smoothed adversaries. For finite hypothesis classes $\mathcal{H}$, existing no-regret algorithms such as Hedge [Freund and Schapire, 1997] and Follow-the-Perturbed-Leader [Kalai and Vempala, 2005] achieve a regret bound of $O(\sqrt{T \ln(\mathcal{H})})$. For a possibly infinite hypothesis class our approach uses a finite set $\mathcal{H}'$ as a *proxy* for $\mathcal{H}$ and only focuses on competing with hypotheses in $\mathcal{H}'$ by running a standard no-regret algorithm on $\mathcal{H}'$. Indeed, in absence of smoothness of $\mathscr{D}$, $\mathcal{H}'$ has to be a good proxy with respect to every distribution or know the adversarial sequence ahead of time, neither of which are possible in the online setting. But when distributions are smooth, $\mathcal{H}'$ that is a good proxy for the uniform distribution can also be a good proxy for all other smooth distributions. We will see that how well a set $\mathcal{H}'$ approximates $\mathcal{H}$ depends on adaptivity

(versus non-adpativity) of the adversary. Our main technical result in Section 3.1 shows that for adaptive adversaries this approximation depends on the size of the $\frac{\sigma}{4\sqrt{T}}$-bracketing cover of $\mathcal{H}$. This results in an algorithm whose regret is sublinear in $T$ and logarithmic in that bracketing number for adaptive adversaries (Theorem 3.3). In comparison, for simpler non-adaptive adversaries this approximation depends on the size of the more traditional $\epsilon$-covers of $\mathcal{H}$, which do not require pointwise approximation of $\mathcal{H}$. This leads to an algorithm against non-adaptive adversaries with an improved regret bound of $\tilde{O}(\sqrt{T \cdot \mathrm{VCDim}(\mathcal{H})})$ (Theorem 3.3).

In Section 3.2, we demonstrate that the bracketing numbers of commonly used hypothesis classes in machine learning are small functions of their VC dimension. We also provide systematic approaches for bounding the bracketing number of complex hypothesis classes in terms of the bracketing number of their simpler building blocks. This shows that for many commonly used hypothesis classes — such as halfspaces, polynomial threshold functions, and polytopes — we can achieve a regret of $\tilde{O}(\sqrt{T \cdot \mathrm{VCDim}(\mathcal{H})})$ even against an adaptive adversary.

## 3.1 Regret Analysis and the Connection to Bracketing Number

In more detail, consider an algorithm $\mathcal{A}$ that uses Hedge on a finite set $\mathcal{H}'$ instead of $\mathcal{H}$. Then,

$$\mathbb{E}[\mathrm{REGRET}(\mathcal{A}, \boldsymbol{\mathcal{D}})] \le O\left(\sqrt{T \ln(|\mathcal{H}'|)}\right) + \mathbb{E}_{\boldsymbol{\mathcal{D}}}\left[\max_{h \in \mathcal{H}} \min_{h' \in \mathcal{H}'} \sum_{t=1}^{T} \mathbb{1}\left(h(x_t) \ne h'(x_t)\right)\right], \quad (1)$$

where the first term is the regret against the best $h' \in \mathcal{H}'$ and the second term captures how well $\mathcal{H}'$ approximates $\mathcal{H}$. A natural choice of $\mathcal{H}'$ is an $\epsilon$-cover of $\mathcal{H}$ with respect to the uniform distribution, for a small $\epsilon$ that will be defined later. This bounds the first term using the fact that there is an $\epsilon$-cover $\mathcal{H}' \subseteq \mathcal{H}$ of size $|\mathcal{H}'| \le (41/\epsilon)^{\mathrm{VCDim}(\mathcal{H})}$. To bound the second term, we need to understand whether there is a hypothesis $h \in \mathcal{H}$ whose value over *an adaptive sequence of $\sigma$-smooth distributions* can be drastically different from the value of its closest (under uniform distribution) proxy $h' \in \mathcal{H}'$. Considering the symmetric difference functions $f_{h,h'} = h\Delta h'$ for functions $h \in \mathcal{H}$ and their corresponding proxies $h' \in \mathcal{H}'$, we need to bound (in expectation) the maximum value an $f_{h,h'}$ can attain over an adaptive sequence of $\sigma$-smooth distributions.

**Non-Adaptive Adversaries.** To develop more insight, let us first consider the case of *non-adaptive* adversaries. In the case of non-adaptive adversaries, $x_t \sim \mathcal{D}_t$ are *independent* of each other, while they are not identically distributed. This independence is the key property that allows us to use the VC dimension of the set of functions $\{f_{h,h'} \mid \forall h \in \mathcal{H} \text{ and the corresponding proxy } h' \in \mathcal{H}'\}$ to establish a uniform convergence property where with high probability every function $f_{h,h'}$ has a value that is close to its expectation — the fact that $x_t$s are not identically distributed can be easily handled because the double sampling and symmetrization trick in VC theory can still be applied as before. Furthermore, $\sigma$-smoothness of the distributions implies that $\mathbb{E}_{\boldsymbol{\mathcal{D}}}[\sum f_{h,h'}(x_t)] \le \sigma^{-1}\mathbb{E}_{\mathcal{U}}[\sum f_{h,h'}(x_t)] \le \epsilon/\sigma$. This leads to the following theorem for non-adaptive adversaries.

**Theorem 3.1** (Non-Adaptive Adversary [Haghtalab, 2018]). *Let $\mathcal{H}$ be a hypothesis class of VC dimension $d$. There is an algorithm such that for any $\boldsymbol{\mathcal{D}}$ that is an non-adaptive sequence of $\sigma$-smooth distributions has regret $\mathbb{E}[\mathrm{REGRET}(\mathcal{A}, \boldsymbol{\mathcal{D}})] \in O\left(\sqrt{dT \ln\left(\frac{T}{\sigma}\right)}\right)$.*

**Adaptive Adversaries.** Moving back to the case of adaptive adversaries, we unfortunately lose this uniform convergence property (see Appendix B for an example). This is due to the fact that now the choice of $\mathcal{D}_t$ can depend on the earlier realization of instances $x_1, \ldots, x_{t-1}$. To see why independence is essential, note that the ubiquitous double sampling and symmetrization techniques used in VC theory require that taking two sets of samples $\mathbf{x}$ and $\mathbf{x}'$ from the process that is generating data, we can swap $x_i$ and $x_i'$ independently of whether $x_j$ and $x_j'$ are swapped for $j \ne i$. When the choice of $\mathcal{D}_t$ depends on $x_1, \ldots, x_{t-1}$ then swapping $x_\tau$ with $x_\tau'$ affects whether $x_t$ and $x_t'$ could even be generated from $\mathcal{D}_t$ for $t > \tau$. In other words, symmetrizing the first $t$ variables generates $2^t$ possible choices for $x^{t+1}$ that exponentially increases the set of samples over which a VC class has to be projected, therefore losing the typical $\sqrt{T \cdot \mathrm{VCDim}(\mathcal{H})}$ regret bound and instead obtaining the trivial regret of $O(T)$. Nevertheless, we show that the earlier ideas for bounding the second term of Equation 1 are still relevant as long as we can side step the need for independence.

Note that $\sigma$-smoothness of the distributions still implies that for a fixed function $f_{h,h'}$ even though $\mathcal{D}_t$ is dependent on the realizations $x_1, \ldots, x_{t-1}$, we still have $\Pr_{x_t \sim \mathcal{D}_t}[f_{h,h'}(x_t)] \leq \epsilon/\sigma$. Indeed, the value of any function $f$ for which $\mathbb{E}_{\mathcal{U}}[f(x)] \leq \epsilon$ can be bounded by the convergence property of an appropriately chosen Bernoulli variable. As we demonstrate in the following lemma, this allows us to bound the expected maximum value of a $f_{h,h'}$ *chosen from a finite set of symmetric differences.* For a proof of this lemma refer to Appendix C.2.

**Lemma 3.2.** *Let $\mathcal{F} : \mathcal{X} \to \{0,1\}$ be any finite class of functions such that $\mathbb{E}_{\mathcal{U}}[f(x)] \leq \epsilon$ for all $f \in \mathcal{F}$, i.e., every function has measure $\epsilon$ over the uniform distribution. Let $\mathcal{D}$ be any adaptive sequence of $T$, $\sigma$-smooth distributions for some $\sigma \geq \epsilon$ such that $T \frac{\epsilon}{\sigma} \geq \sqrt{\ln(|\mathcal{F}|)}$. We have that*

$$\mathbb{E}_{\mathbf{x} \sim \mathcal{D}} \left[ \max_{f \in \mathcal{F}} \sum_{t=1}^{T} f(x_t) \right] \leq O\left( T \frac{\epsilon}{\sigma} \sqrt{\ln(|\mathcal{F}|)} \right).$$

The set of symmetric differences $\mathcal{G} = \{ f_{h,h'} \mid \forall h \in \mathcal{H} \text{ and the corresponding proxy } h' \in \mathcal{H}' \}$ we work with is of course infinitely large. Therefore, to apply Lemma 3.2 we have to approximate $\mathcal{G}$ with a finite set $\mathcal{F}$ such that

$$\mathbb{E}_{\mathbf{x} \sim \mathcal{D}} \left[ \max_{f_{h,h'} \in \mathcal{G}} \sum_{t=1}^{T} f_{h,h'}(x_t) \right] \lesssim \mathbb{E}_{\mathbf{x} \sim \mathcal{D}} \left[ \max_{f \in \mathcal{F}} \sum_{t=1}^{T} f(x_t) \right]. \tag{2}$$

What should this set $\mathcal{F}$ be? Note that choosing $\mathcal{F}$ that is an $\epsilon$-cover of $\mathcal{G}$ under the uniform distribution is an ineffective attempt plagued by the same lack of independence that we are trying to side step. In fact, while all functions $f_{h,h'}$ are $\epsilon$ close to the constant $0$ functions with respect to the uniform distribution, they are activated on different parts of the domain. So it is not clear that an adaptive adversary, who can see the earlier realizations of instances, cannot ensure that one of these regions will receive a large number realized instances. But a second look at Equation 2 suffices to see that this is precisely what we can obtain if $\mathcal{F}$ were to be the set of (upper) functions in an $\epsilon$-bracketing of $\mathcal{G}$. That is, for every function $f_{h,h'} \in \mathcal{G}$ there is a function $f \in \mathcal{F}$ such that $f_{h,h'} \preceq f$. This proves Equation 2 with an exact inequality using the fact that pointwise approximation $f_{h,h'} \preceq f$ implies that the value of $f_{h,h'}$ is bounded by that of $f$ for any set of instances $x_1, \ldots, x_T$ that could be generated by $\mathcal{D}$. Furthermore, functions in $\mathcal{G}$ are within $\epsilon$ of the constant $0$ function over the uniform distribution, so $\mathcal{F}$ meets the criteria of Lemma 3.2 with the property that for all $f \in \mathcal{F}$, $\mathbb{E}_{\mathcal{U}}[f(x)] \leq \epsilon$. It remains to bound the size of class $|\mathcal{F}|$ in terms of the bracketing number of $\mathcal{H}$. This can be done by showing that the bracketing number of class $\mathcal{G}$, that is the class of all symmetric differences in $\mathcal{H}$, is approximately bounded by the same bracketing number of $\mathcal{H}$ (See Theorem 3.7 for more details). Putting these all together we get the following regret bound against smoothed adaptive adversaries.

**Theorem 3.3** (Adaptive Adversary). *Let $\mathcal{H}$ be a hypothesis class over domain $\mathcal{X}$, whose $\epsilon$-bracketing number with respect to the uniform distribution over $\mathcal{X}$ is denoted by $\mathcal{N}_{[]}(\mathcal{H}, \mathcal{U}, \epsilon)$. There is an algorithm such that for any $\mathcal{D}$ that is an adaptive sequence of $\sigma$-smooth distributions has regret*

$$\mathbb{E}[\text{REGRET}(\mathcal{A}, \mathcal{D})] \in O\left( \sqrt{T \ln\left( \mathcal{N}_{[]}\left( \mathcal{H}, \mathcal{U}, \frac{\sigma}{4\sqrt{T}} \right) \right)} \right).$$

## 3.2 Hypothesis Classes with Small Bracketing Numbers.

In this section, we analyze bracketing numbers of some commonly used hypothesis classes in machine learning. We start by reviewing the bracketing number of halfspaces and provide two systematic approaches for extending this bound to other commonly used hypothesis classes. Our first approach bounds the bracketing number of any class using the dimension of the space needed to embed it as halfspaces. Our second approach shows that $k$-fold operations on any hypothesis class, such as taking the class of intersections or unions of all $k$ hypotheses in a class, only mildly increase the bracketing number. Combining these two techniques allows us to bound the bracketing number of commonly used classifiers such as halfspaces, polytopes, polynomial threshold functions, etc.

The connection between bracketing number and VC theory has been explored in recent works. Adams and Nobel [2010, 2012] showed that finite VC dimension class also have finite $\epsilon$-bracketing number but Alon et al. [1987] (see van Handel [2013] for a modern presentation) showed the dependence on $1/\epsilon$ can be arbitrarily bad. Since Theorem 3.3 depends on the growth rate of bracketing numbers, we work with classes for which we can obtain $\epsilon$-bracketing numbers with reasonable growth rate, those that are close to the size of standard $\epsilon$-covers.

**Theorem 3.4** (Braverman et al. [2019])**.** *Let $\mathcal{H}$ be the class of halfspaces over $\mathbb{R}^d$. For any $\epsilon > 0$ and any measure $\mu$ over $\mathbb{R}^d$, $\mathcal{N}_{[]}(\mathcal{H}, \mu, \epsilon) \leq \left(\frac{d}{\epsilon}\right)^{O(d)}$.*

Our first technique uses this property of halfspaces to bound the bracketing number of any hypothesis class as a function of the dimension of the spaces needed to embed this class as halfpsaces.

**Definition 3.5** (Embeddable Classes)**.** *Let $\mathcal{G}$ be a hypothesis class on $\mathcal{X}$. We say that $\mathcal{G}$ is embeddable as halfspaces in $m$ dimensions if there exists a map $\psi : \mathcal{X} \to \mathbb{R}^m$ such that for any $g \in \mathcal{G}$, there is a linear threshold function $h$ such $g = h \circ \psi$.*

**Theorem 3.6** (Bracketing Number of Embeddable Classes)**.** *Let $\mathcal{G}$ be a hypothesis class embeddable as halfspaces in $m$ dimensions. Then, for any measure $\nu$, $\mathcal{N}_{[]}(\mathcal{G}, \nu, \epsilon) \leq \left(\frac{m}{\epsilon}\right)^{O(m)}$.*

Our second technique shows that combining $k$ classes, by respectively taking intersections or unions of any $k$ functions from them, only mildly increases their bracketing number.

**Theorem 3.7** (Bracketing Number of $k$-fold Operations)**.** *Let $\mathcal{F}_1, \ldots, \mathcal{F}_k$ be $k$ hypothesis classes. Let $\mathcal{F}_1 \cdot \mathcal{F}_2 \cdots \mathcal{F}_k$ and $\mathcal{F}_1 + \mathcal{F}_2 + \cdots + \mathcal{F}_k$ be the class of all hypotheses that are intersections and unions of $k$ functions $f_i \in \mathcal{F}_i$, respectively. Then,*

$$\mathcal{N}_{[]}(\mathcal{F}_1 \cdot \mathcal{F}_2 \cdots \mathcal{F}_k, \mu, k\epsilon) \leq \prod_{i \in [k]} \mathcal{N}_{[]}(\mathcal{F}_i, \mu, \epsilon)$$

*and*

$$\mathcal{N}_{[]}(\mathcal{F}_1 + \mathcal{F}_2 + \cdots + \mathcal{F}_k, \mu, k\epsilon) \leq \prod_{i \in [k]} \mathcal{N}_{[]}(\mathcal{F}_i, \mu, \epsilon) .$$

*For any hypothesis class $\mathcal{F}$ and $\mathcal{G} = \{f \Delta f' \mid \text{for all} f, f' \in \mathcal{F}\}$, $\mathcal{N}_{[]}(\mathcal{G}, \mu, 4\epsilon) \leq \left(\mathcal{N}_{[]}(\mathcal{F}, \mu, \epsilon)\right)^4$.*

We now use our techniques for bounding the bracketing number of complex classes by the bracketing number of their simpler building blocks to show that online learning with an adaptive adversary on a class of halfspaces, polytopes, and polynomial threshold functions has $\tilde{O}(\sqrt{T \, \text{VCDim}(\mathcal{H})})$ regret.

**Corollary 3.8.** *Consider instance space $\mathcal{X} = \mathbb{R}^n$ and let $\mu$ be an arbitrary measure on $\mathcal{X}$. Let $\mathcal{P}^{n,d}$ be the class of $d$-degree polynomial thresholds and $\mathcal{Q}^{n,k}$ be the class $k$-polytopes in $\mathbb{R}^n$. Then,*

$$\mathcal{N}_{[]}\left(\mathcal{P}^{n,d}, \mu, \epsilon\right) \leq \exp\left(c_1 n^d \ln\left(n^d/\epsilon\right)\right) \text{ and } \mathcal{N}_{[]}\left(\mathcal{Q}^{n,k}, \mu, \epsilon\right) \leq \exp\left(c_2 nk \ln\left(\frac{nk}{\epsilon}\right)\right),$$

*for some constants $c_1$ and $c_2$. Furthermore, there is an online algorithm whose regret against an adaptive $\sigma$-smoothed adversary on the class $\mathcal{P}^{n,d}$ and $\mathcal{Q}^{n,k}$ is respectively $\tilde{O}(\sqrt{T \cdot \text{VCDim}(\mathcal{P}^{n,d}) \ln(1/\sigma)})$ and $\tilde{O}(\sqrt{T \cdot \text{VCDim}(\mathcal{Q}^{n,k}) \ln(1/\sigma)})$.*

## 4 Differential Privacy

In this section, we consider smoothed analysis of differentially private learning in *query answering* and *data release* paradigms. We primarily focus on $(\sigma, 0)$-smooth distributions and defer the general case of $(\sigma, \chi)$-smooth distributions to Appendix G. For finite query classes $\mathcal{Q}$ and small domains, existing differentially private mechanisms achieve an error bound that depends on $\ln(|\mathcal{Q}|)$ and $\ln(|\mathcal{X}|)$. We leverage smoothness of data sets to improve these dependencies to $\text{VCDim}(\mathcal{Q})$ and $\ln(1/\sigma)$.

**An Existing Algorithm.** Hardt et al. [2012] introduced a practical algorithm for data release, called Multiplicative Weights Exponential Mechanism (MWEM). This algorithm works for a finite query class $\mathcal{Q}$ over a finite domain $\mathcal{X}$. Given an data set $B$ and its corresponding empirical distribution $\mathcal{D}_B$, MWEM iteratively builds distributions $\mathcal{D}_t$ for $t \in [T]$, starting from $\mathcal{D}_1 = \mathcal{U}$ that is the uniform distribution over $\mathcal{X}$. At stage $t$, the algorithm picks a $q_t \in \mathcal{Q}$ that approximately maximizes the error $|q_t(\mathcal{D}_{t-1}) - q_t(\mathcal{D}_B)|$ using a differentially private mechanism (Exponential mechanism). Then data set $\mathcal{D}_{t-1}$ is updated using the multiplicative weights update rule $\mathcal{D}_t(x) \propto \mathcal{D}_{t-1}(x) \exp\left(q_t(x)(m_t - q_t(\mathcal{D}_{t-1}))/2\right)$ where $m_t$ is a differentially private estimate (via Laplace mechanism) for the value $q_t(\mathcal{D}_B)$. The output of the mechanism is a data set $\overline{\mathcal{D}} = \frac{1}{T} \sum_{t \in [T]} \mathcal{D}_t$. The formal guarantees of the algorithm are as follows.

**Theorem 4.1** (Hardt et al. [2012]). *For any data set $B$ of size $n$, a finite query class $\mathcal{Q}$, $T \in \mathbb{N}$ and $\epsilon > 0$, MWEM is $\epsilon$-differentially private and with probability at least $1 - {}^{2T}/_{|\mathcal{Q}|}$ produces a distribution $\overline{\mathcal{D}}$ over $\mathcal{X}$ such that $\max_{q \in \mathcal{Q}} \left\{ \left| q\left(\overline{\mathcal{D}}\right) - q\left(\mathcal{D}_B\right) \right| \right\} \leq 2\sqrt{\frac{\log|\mathcal{X}|}{T}} + \frac{10T \log|\mathcal{Q}|}{\epsilon n}$.*

The analysis of MWEM keeps track of the KL divergence $\mathrm{D_{KL}}\left(\mathcal{D}_B \| \mathcal{D}_t\right)$ and shows that at time $t$ this value decreases by approximately the error of query $q_t$. At a high level, $\mathrm{D_{KL}}\left(\mathcal{D}_B \| \mathcal{D}_1\right) \leq \ln(|\mathcal{X}|)$. Moreover, KL divergence of any two distributions is non-negative. Therefore, error of any query $q \in \mathcal{Q}$ after $T$ steps follows the above bound.

**Query Answering.** To design a private query answering algorithm for a query class $\mathcal{Q}$ without direct dependence on $\ln(|\mathcal{Q}|)$ and $\ln(|\mathcal{X}|)$ we leverage smoothness of distributions. Our algorithm called the Smooth Multiplicative Weight Exponential Mechanism (Smooth MWEM), given an infinite set of queries $\mathcal{Q}$, considers a $\gamma$-cover $\mathcal{Q}'$ under the uniform distribution. Then, it runs the MWEM algorithm with $\mathcal{Q}'$ as the query set and constructs an empirical distribution $\overline{\mathcal{D}}$. Finally, upon being requested an answer to a query $q \in \mathcal{Q}$, it responds with $q'(\overline{\mathcal{D}})$, where $q' \in \mathcal{Q}'$ is the closest query to $q$ under the uniform distribution. This algorithm is presented in Appendix E. Note that $\mathcal{Q}'$ does not depend on the data set $B$. This is the key property that enables us to work with a finite $\gamma$-cover of $\mathcal{Q}$ and extend the privacy guarantees of MWEM to infinite query classes. In comparison, constructing a $\gamma$-cover of $\mathcal{Q}$ with respect to the empirical distribution $\mathcal{D}_B$ uses private information.

Let us now analyze the error of our algorithm and outline the reasons it does not directly depend on $\ln(|\mathcal{Q}|)$ and $\ln(|\mathcal{X}|)$. Recall that from the $(\sigma, 0)$-smoothness, there is a distribution $\overline{\mathcal{D}_B}$ that is $\sigma$-smooth and $q(\mathcal{D}_B) = q\left(\overline{\mathcal{D}_B}\right)$ for all $q \in \mathcal{Q}$. Furthermore, $\mathcal{Q}'$ can be taken to be a subset of $\mathcal{Q}$ and thus $B$ is $(\sigma, 0)$-smooth with respect to $\mathcal{Q}'$. The approximation of $\mathcal{Q}$ by a $\gamma$-cover introduces error in addition to the error of Theorem 4.1. This error is given by $|q(\mathcal{D}_B) - q'(\mathcal{D}_B)| \leq 2 \cdot \Pr_{\mathcal{U}}\left[q'(x) \neq q(x)\right] \sigma^{-1} \leq 2\gamma/\sigma$. Note that $|\mathcal{Q}'| \leq (41/\gamma)^{\mathrm{VCDim}(\mathcal{Q})}$, therefore, this removes the error dependence on the size of the query set $\mathcal{Q}$ while adding a small error of $2\gamma/\sigma$. Furthermore, Theorem 4.1 dependence on $\ln(|\mathcal{X}|)$ is due to the fact that for a worst-case (non-smooth) data set $B$, $\mathrm{D_{KL}}(\mathcal{D}_B \| \mathcal{U})$ can be as high as $\ln(|\mathcal{X}|)$. For a $(\sigma, 0)$-smooth data set, however, $\mathrm{D_{KL}}(\overline{\mathcal{D}_B} \| \mathcal{U}) \leq \ln(1/\sigma)$. This allows for faster error convergence. Applying these ideas together and setting $\gamma = \sigma/2n$ gives us the following theorem whose proof is deferred to Appendix E.

**Theorem 4.2.** *For any $(\sigma, 0)$-smooth dataset $B$ of size $n$, a query class $\mathcal{Q}$ with VC dimension $d$, $T \in \mathbb{N}$ and $\epsilon > 0$, Smooth Multiplicative Weights Exponential Mechanism is $\epsilon$-differentially private and with probability at least $1 - 2T \left(\gamma/_{41}\right)^{\mathrm{VCDim}(\mathcal{Q})}$, calculates values $v_q$ for all $q \in \mathcal{Q}$ such that*

$$\max_{q \in \mathcal{Q}} \left\{ |v_q - q(\mathcal{D}_B)| \right\} \leq \frac{1}{n} + 2\sqrt{\frac{\log\left(1/\sigma\right)}{T}} + \frac{10Td \log\left(2n/\sigma\right)}{\epsilon n}.$$

**Data Release.** Above we described a procedure for query answering that relied on the construction of a data set. One could ask whether this leads to a solution to the data release problem as well. An immediate, but ineffective, idea is to output distribution $\overline{\mathcal{D}}$ constructed by our algorithm in the previous section. The problem with this approach is that while $q'(\overline{\mathcal{D}}) \approx q'(\mathcal{D}_B)$ for all queries in the cover $\mathcal{Q}'$, there can be queries $q \in \mathcal{Q} \setminus \mathcal{Q}'$ for which $\left| q(\overline{\mathcal{D}}) - q(\mathcal{D}_B) \right|$ is quite large. This is due to the fact that even though $B$ is $(\sigma, 0)$-smooth (and $\overline{\mathcal{D}_B}$ is $\sigma$-smooth), the repeated application of multiplicative update rule may result in distribution $\overline{\mathcal{D}}$ that is far from being smooth.

To address this challenge, we introduce Projected Smooth Multiplicative Weight Exponential Mechanism (Projected Smooth MWEM) that ensures that $\mathcal{D}_t$ is also $\sigma$-smooth by projecting it on the convex set of all $\sigma$-smooth distributions. More formally, let $\mathcal{K}$ be the polytope of all $\sigma$-smooth distributions over $\mathcal{X}$ and let $\tilde{\mathcal{D}}_t$ be the outcome of the multiplicative update rule of Hardt et al. [2012] at time $t$. Then, Projected Smooth MWEM mechanism uses $\mathcal{D}_t = \arg\min_{\mathcal{D} \in \mathcal{K}} \mathrm{D_{KL}}(\mathcal{D} \| \tilde{\mathcal{D}}_t)$. To ensure that these projections do not negate the progress made so far, measured by the decrease in KL divergence, we note that for any $\overline{\mathcal{D}_B} \in \mathcal{K}$ and any $\tilde{\mathcal{D}}_t$, we have $\mathrm{D_{KL}}(\overline{\mathcal{D}_B} \| \tilde{\mathcal{D}}_t) \geq \mathrm{D_{KL}}(\overline{\mathcal{D}_B} \| \mathcal{D}_t) + \mathrm{D_{KL}}(\mathcal{D}_t \| \tilde{\mathcal{D}}_t)$. That is, as measured by the decrease in KL divergence, the improvement with respect to $\mathcal{D}_t$ can only be greater than that of $\tilde{\mathcal{D}}_t$. Optimizing parameters $T$ and $\gamma$, we obtain the following guarantees. See Appendix F for more details on Projected Smooth MWEM mechanism and its analysis.

**Theorem 4.3** (Smooth Data Release). *Let $B$ be a $\sigma$-smooth database with $n$ data points. For any $\epsilon, \delta > 0$ and any query set $\mathcal{Q}$ with VC dimension $d$, Projected Smooth Multiplicative Weight*

*Exponential Mechanism* is $(\epsilon, \delta)$ *differentially private and with probability at least* $1 - 1/poly\,(n/\sigma)^d$ *its outcome* $\overline{\mathcal{D}}$ *satisfies*

$$\max_{q \in \mathcal{Q}} \left\{ \left| q\left(\overline{\mathcal{D}}\right) - q\left(\mathcal{D}_B\right) \right| \right\} \leq O\left( \sqrt{\frac{d}{\epsilon n} \log^{\frac{1}{2}}\left(\frac{1}{\sigma}\right) \log\left(\frac{n}{\sigma}\right) \log\left(\frac{1}{\delta}\right)} \right).$$

## 5   Conclusions and Open Problems

Our work introduces a framework for smoothed analysis of online and private learning and obtain regret and error bounds that depend only on the VC dimension and the bracketing number of a hypothesis class and are independent of the domain size and Littlestone dimension.

Our work leads to several interesting questions for future work. The first is to characterize learnability in the smoothed setting — via matching lower bounds — in terms of a combinatorial quantity, e.g., bracketing number. In Appendix D, we discuss *sign rank* and its connection to bracketing number as a promising candidate for this characterization. A related question is whether there are finite VC dimension classes that cannot be learned in presence of smoothed adaptive adversaries.

Let us end this paper by noting that the Littlestone dimension plays a key role in characterizing learnability and algorithm design in the worst-case for several socially and practically important constraints [Ben-David et al., 2009, Alon et al., 1987]. It is essential then to develop models that can bypass Littlestone impossibility results and provide rigorous guidance in achieving these constraints in practical settings.

## Broader Impact

Like many theoretical machine learning papers, this paper's main focus is on the mathematical challenges and contributions to the field. However, as robustness and privacy are two of the most important practical and societal challenges machine learning is facing, our work also has broader implications on the deployments of these techniques.

The theoretical impossibility results in online learning and differential privacy have been barriers to developing robust and private learning methods that work well on day-to-day applications and have rigorous guarantees — e.g., the impossibility result for privacy implies that error guarantees of private learning methods only work when data sets are *infinitely* large. Our work provides a framework to side step these impossibility results. Our online and differentially private algorithms perform as well as their offline and non-private counterparts on real-life data and are backed by rigorous theoretical guarantees.

## Acknowledgements

This work was partially supported by the NSF under CCF-1813188, the ARO under W911NF1910294 and a JP Morgan Chase Faculty Fellowship.

## Footnotes

[1] Along the way, we also demonstrate a stronger regret bound for the simpler case of non-adaptive adversaries, for which each distribution $\mathcal{D}_t$ is independent of the realized inputs in previous time steps.

[2]In differentially private learning, queries are the error function of hypotheses and take as input a pair $(x, y)$.

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
