[Supplementary Material 1 · Full_Paper.pdf]

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

# A Additional Related Work

Analogous models of smoothed online learning have been explored in prior work. Rakhlin et al. [2011] consider online learning when the adversary is constrained in several ways and work with a notion of sequential Rademacher complexity for analyzing the regret. In particular, they study a related notion of smoothed adversary and show that one can learn thresholds with regret of $O(\sqrt{T})$ in presence of smoothed adversaries. Gupta and Roughgarden [2017] consider smoothed online learning in the context online algorithm design. They show that while optimizing parameterized greedy heuristics for Maximum Weight Independent Set imposes linear regret in the worst-case, in presence of smoothing this problem can be learned with sublinear regret (as long they allow per-step runtime that grows with $T$). Cohen-Addad and Kanade [2017] consider the same problem with an emphasis on the per-step runtime being logarithmic in $T$. They show that piecewise constant functions over the interval $[0, 1]$ can be learned efficiently within regret of $O(\sqrt{T})$ against a *non-adaptive* smooth adversary. Our work differs from these by upper bounding the regret using a combinatorial dimension of the hypothesis class and demonstrating techniques that generalize to large class of problems in presence of *adaptive* adversaries.

In another related work, Balcan et al. [2018] introduce a notion of dispersion in online optimization (where the learner picks an instance and the adversary picks a function) that is a constraint on the number of discontinuities in the adversarial sequence of functions. They show that online optimization can be done efficiently under certain assumptions. Moreover, they show that sequences generated by *non-adaptive* smooth adversaries in one dimension satisfy dispersion. In comparison, our main results in online learning consider the more powerful adaptive adversaries.

Smoothed analysis is also used in a number of other online settings. In the setting of linear contextual bandits, Kannan et al. [2018] use smoothed analysis to show that the greedy algorithm achieves sublinear regret even though in the worst case it can have linear regret. Raghavan et al. [2018] work in a Bayesian version of the same setting and achieve improved regret bounds for the greedy algorithm. Since several algorithms are known to have sublinear regret in the linear contextual bandit setting even in the worst-case, the main contribution of these papers is to show that the simple and practical greedy algorithm has much better regret guarantees than in the worst-case. In comparison, we work with a setting where no algorithm can achieve sublinear regret in the worst-case.

Smoothed analysis has also been considered in the context of differential privacy. Hardt and Rothblum [2010] consider differential privacy in the interactive setting, where the queries arrive online. They analyze a multiplicative weights based algorithm whose running time and error they show can be vastly improved in the presence of smoothness. Some of our techniques for query answering and data release are inspired by that line of work. Balcan et al. [2018] also differential privacy in presence of dispersion and analyze the gaurantees of the exponential mechanism.

Generally, our work is also related to a line of work on online learning in presence of additional assumptions resembling properties exhibited by real life data. Rakhlin and Sridharan [2013] consider settings where additional information in terms of an estimator for future instances is available to the learner. They achieve regret bounds that are in terms of the path length of these estimators and can beat $\Omega(\sqrt{T})$ if the estimators are accurate. Dekel et al. [2017] also considers the importance of incorporating side information in the online learning framework and show that regrets of $O(\log(T))$ in online linear optimization maybe possible when the learner knows a vector that is weakly correlated with the future instances.

More broadly, our work is among a growing line of work on beyond the worst-case analysis of algorithms [Roughgarden, 2020] that considers the design and analysis of algorithms on instances that satisfy properties demonstrated by real-world applications. Examples of this in theoretical machine learning mostly include improved runtime and approximation guarantees of numerous supervised (e.g., [Kalai et al., 2009, Kalai and Teng, 2008, Awasthi et al., 2016, Diakonikolas et al., 2019]), and unsupervised settings (e.g., [Bilu and Linial, 2012, Balcan et al., 2020, 2013, Arora et al., 2012, Bhaskara et al., 2019, Vijayaraghavan et al., 2017, Makarychev et al., 2014, Ostrovsky et al., 2013, Hardt and Roth, 2013]).

# B  Lack of Uniform Convergence with Adaptive Adversaries

The following example for showing lack of uniform convergence over adaptive sequences is due to
Haghtalab [2018] and is included here for completeness.

Let $\mathcal{X} = [0, 1]$ and $\mathcal{G} = \{g_b(x) = \mathbb{I}(x \geq b) \mid \forall b \in [0, 1]\}$ be the set of one-dimensional thresholds.
Let the distribution of the noise $\eta_i$ be the uniform distribution on $(-1/4, 1/4)$. Let $x_1 = 1/2$ and
$x_2 = x_3 = \cdots = x_T = 1/4$ if $\eta_1 \leq 0$ while $x_2 = x_3 = \cdots = x_T = 3/4$ otherwise. In this case, we
do not achieve concentration for any value of $T$, as

$$\frac{1}{T} \sum_{t=1}^{T} g_{0.5}(x_t + \eta_t) = \begin{cases} 0 & \text{w.p. } 1/2 \\ 1 & \text{w.p. } 1/2 \end{cases} \qquad and \qquad \mathbb{E}\left[\frac{1}{T} \sum_{t=1}^{T} g_{0.5}(x_t + \eta_t)\right] = \frac{1}{2}.$$

# C  Proofs from Section 3

## C.1  Algorithm and its Running Time

While our main focus is to provide sublinear regret bounds for smoothed online learning our analysis
also provides an algorithmic solution describe below.

---

**Algorithm 1:** Smooth Online Learning

**Input:** Instance Space $\mathcal{X}$, Hypothesis Class $\mathcal{H}$, Smoothness parmeter $\sigma$, Time horizon $T$

*Cover Construction:* Compute $\mathcal{H}' \subseteq \mathcal{H}$ that is a $\gamma$-cover of $\mathcal{H}$ with respect to the uniform
  distribution on $\mathcal{X}$ for $\gamma = \frac{\sigma}{4\sqrt{T}}$.

**for** $t = 1 \ldots T$ **do**
  Use a standard online learning algorithm, such as Hedge, on $\mathcal{H}'$ to pick an $h_t$, where the
    history of the play is $\{s_\tau\}_{\tau < t}$ and $\{h_\tau\}_{\tau < t}$
  Receive $s_t = (x_t, y_t)$ and suffer loss $\text{err}_{s_t}(h_t)$.
**end**

---

The running time of the algorithm comprises of the initial construction of $\mathcal{H}'$ and then running a
standard online learning algorithm on $\mathcal{H}'$.

Standard online learning algorithms such as Hedge and FTPL take time polynomial in the size of the
cover since in standard implementations they maintain a state corresponding to each hypothesis in
$\mathcal{H}'$. In our setting, the size of the cover is $(41\sqrt{T}/\sigma)^d$.

The time required to construct a cover depends on the access we have to the class. One method is to
randomly sample a set $S$ with $m = O(\text{VCDim}(\mathcal{H})T/\sigma^2)$ points from the domain uniformly and
construct all possible labelings on this set induced by the class. The number of labellings of $S$ is
bounded by $O(m^{\text{VCDim}(\mathcal{H})})$ by the Sauer–Shelah lemma. The cover is constructed by then finding
functions in the class $\mathcal{H}$ that are consistent with each of these labellings. This requires us to be able
to find an element in the class consistent with a given labeling, which can be done by a "consistency"
oracle. Naively, the above makes $2^m$ calls to the consistency oracle, one for each possible labeling of
$S$.

The above analysis and runtime can be improved in several ways. First, $\mathcal{H}'$ can be constructed in
time $O(m^{\text{VCDim}(\mathcal{H})})$ rather than $2^m$. This can be done by constructing the cover in a hierarchical
fashion, where the root includes the unlabeled set $S$ and at every level one additional instance in $S$
is labeled by $+1$ or $-1$. At each node, the consistency oracle will return a function $h \in \mathcal{H}$ that is
consistent with the labels so far or state that none exists. Nodes for which no consistent hypothesis so
far exists are pruned and will not expand in the next level. Since the total number of leaves is the
number of ways in which $S$ can be labeled by $\mathcal{H}$, i.e., $O(m^d)$, the number of calls to the consistency
oracle is $O(m^d)$ as well. The runtime of standard online learning algorithms can also be improved
significantly when an empirical risk minimization oracle is available to the learner, in which case
a runtime of $O(\sqrt{|\mathcal{H}'|})$ for general classes [Hazan and Koren, 2016] or even $\text{polylog}(|\mathcal{H}'|)$ for
structured classes [Dudík et al., 2017] is possible.

## C.2  Proof of Lemma 3.2

At a high level, note that any $f \in \mathcal{F}$ has measure at most $\epsilon/\sigma$ on any (even adaptively chosen) $\sigma$-smooth distribution. Therefore, for any fixed $f$, $\mathbb{E}_{\mathscr{D}}[\sum_{t=1}^{T} f(x_t)] \leq T\epsilon/\sigma$. To achieve this bound over all $f \in \mathcal{F}$, we take a union bound over all such functions.

More formally, for any $s$

$$\exp\left(s \mathop{\mathbb{E}}_{\mathscr{D}}\left[\max_{f \in \mathcal{F}} \sum_{t=1}^{T} f(x_t)\right]\right) \leq \mathop{\mathbb{E}}_{\mathscr{D}}\left[\exp\left(s \max_{f \in \mathcal{F}} \sum_{t=1}^{T} f(x_t)\right)\right] \qquad \text{(Jensen's inequqlity)}$$

$$\leq \mathop{\mathbb{E}}_{\mathscr{D}}\left[\max_{f \in \mathcal{F}} \exp\left(s \sum_{t=}^{T} f(x_t)\right)\right] \qquad \text{(Monotonicity of } \exp\text{)}$$

$$\leq \sum_{f \in \mathcal{F}} \mathop{\mathbb{E}}_{\mathscr{D}}\left[\exp\left(s \sum_{t=1}^{T} f(x_t)\right)\right]. \tag{3}$$

Consider a fixed $f \in \mathcal{F}$. Note that even when the choice of a $\sigma$-smoothed distribution $\mathcal{D}$ depends on earlier realizations of $x_1, \ldots, x_{i-1}$, $\Pr_{x_i \sim \mathcal{D}}[f(x_i)] \leq \frac{\epsilon}{\sigma}$. Therefore, $\sum_{t=1}^{T} f(x_t)$ for $\mathbf{x} \sim \mathscr{D}$ is stochastically dominated by that of a binomial distribution $Bin(T, \epsilon/\sigma)$. Note that $\exp(\cdot)$ is a monotonically increasing functions and let $p = \epsilon/\sigma$. We have

$$\mathop{\mathbb{E}}_{\mathscr{D}}\left[\exp\left(s \sum_{t=1}^{T} f(x_t)\right)\right] \leq \sum_{v=0}^{T} \exp(sv) \binom{T}{v} p^v (1-p)^{T-v} = \left(p(\exp(s) - 1) + 1\right)^T. \tag{4}$$

Combining Equations (3) and (4) and noting that $\ln(1 + x) \leq x$, we have

$$\mathop{\mathbb{E}}_{\mathscr{D}}\left[\max_{f \in \mathcal{F}} \sum_{t=1}^{T} f(x_t)\right] \leq \frac{\ln(|\mathcal{F}|) + Tp\left(\exp(s) - 1\right)}{s}.$$

Let $s = \sqrt{\ln(|\mathcal{F}|)/Tp}$. Note that because $s \in (0, 1)$, we have $\exp(s) \leq 1 + 2s$. Hence, by replacing $s$ in the above inequality we have

$$\mathop{\mathbb{E}}_{\mathscr{D}}\left[\max_{f \in \mathcal{F}} \sum_{t=1}^{T} f(x_t)\right] \in O\left(Tp\sqrt{\ln(|\mathcal{F}|)}\right).$$

## C.3  Proof of Theorem 3.3

Consider any hypothesis class $\mathcal{H}'$ and an algorithm that is no-regret with respect to any adaptive adversary on hypotheses in $\mathcal{H}'$. It is not hard to see that

$$\mathbb{E}[\text{REGRET}(\mathcal{A}, \mathscr{D})] = \mathop{\mathbb{E}}_{\mathbf{s} \sim \mathscr{D}}\left[\sum_{t=1}^{T} \text{err}_{s_t}(h_t) - \min_{h \in \mathcal{H}} \text{err}_{s_t}(h_t)\right]$$

$$\leq \mathop{\mathbb{E}}_{\mathbf{s} \sim \mathscr{D}}\left[\sum_{t=1}^{T} \text{err}_{s_t}(h_t) - \min_{h \in \mathcal{H}'} \sum_{t=1}^{T} \text{err}_{s_t}(h)\right] + \mathop{\mathbb{E}}_{\mathbf{s} \sim \mathscr{D}}\left[\min_{h' \in \mathcal{H}'} \sum_{t=1}^{T} \text{err}_{s_t}(h') - \min \sum_{t=1}^{T} \text{err}_{s_t}(h)\right]$$

$$\leq O\left(\sqrt{T \ln(|\mathcal{H}'|)}\right) + \mathop{\mathbb{E}}_{\mathscr{D}}\left[\max_{h \in \mathcal{H}} \min_{h' \in \mathcal{H}'} \sum_{t=1}^{T} \mathbb{1}\left(h(x_t) \neq h'(x_t)\right)\right]. \tag{5}$$

Therefore, it is sufficient to choose an $\mathcal{H}'$ of moderate size such that every function $h \in \mathcal{H}$ has a proxy $h' \in \mathcal{H}'$ even when these functions are evaluated on instances drawn from a *non-iid and adaptive sequence of smooth distributions*. We next describe the choice of $\mathcal{H}'$.

Let $\mathcal{H}'$ be a $\frac{\epsilon}{2}$-net of $\mathcal{H}$ with respect to the uniform distribution $\mathcal{U}$, for an $\epsilon$ that we will determine later. Note that any $\epsilon$-bracket with respect to $\mathcal{U}$ is also an $\epsilon$-net, so $|\mathcal{H}'| \leq \mathcal{N}_{[]}(\mathcal{H}, \mathcal{U}, \epsilon/2)$.[3] Let $\mathcal{G}$ be the set of symmetric differences between $h \in \mathcal{H}$ and its closest proxy $h' \in \mathcal{H}'$, that is,

$$\mathcal{G} = \{g_{h,h'}(x) = \mathbb{1}(h(x) \neq h'(x)) \mid \forall h \in \mathcal{H} \text{ and } h' \in \mathcal{H}', \text{ s.t. } \mathop{\mathbb{E}}_{\mathcal{U}}[g_{h,h'}(x)] \leq \epsilon/2\}.$$

Note that because $\mathcal{G}$ is a subset of all the symmetric differences of two functions in $\mathcal{H}$, by Theorem 3.7 its bracketing number is bounded as follows.

$$\mathcal{N}_{[]}(\mathcal{G},\mathcal{U},\epsilon/2) \leq \left(\mathcal{N}_{[]}(\mathcal{H},\mathcal{U},\epsilon/4)\right)^4. \tag{6}$$

Let $\mathcal{B}(\mathcal{G})$ be the set of upper $\epsilon/2$-brackets of $\mathcal{G}$ with respect to $\mathcal{U}$, i.e., for all $g \in G$, there is $b \in \mathcal{B}(\mathcal{G})$ such that for all $x \in \mathcal{X}$, $g(x) \leq b(x)$ and $\mathbb{E}_{\mathcal{U}}[b(x) - g(x)] \leq \epsilon/2$. Note that

$$\mathbb{E}_{\mathcal{D}}\left[\max_{h\in\mathcal{H}}\min_{h'\in\mathcal{H}'}\sum_{t=1}^{T}\mathbb{1}\left(h(x_t) \neq h'(x_t)\right)\right] = \mathbb{E}_{\mathcal{D}}\left[\max_{g\in\mathcal{G}}\sum_{t=1}^{T}g(x_t)\right] \leq \mathbb{E}_{\mathcal{D}}\left[\max_{b\in\mathcal{B}(\mathcal{G})}\sum_{t=1}^{T}b(x_t)\right],$$

where the last transition is by the fact that $\mathcal{B}(\mathcal{G})$ includes all upper brackets of $\mathcal{G}$.

We now note that $\mathcal{B}(\mathcal{G})$ meets the conditions Lemma 3.2, namely because all $g \in \mathcal{G}$ have measure at most $\epsilon/2$ over $\mathcal{U}$ and $\mathcal{B}(\mathcal{G})$ is the set of $\epsilon/2$-upper brackets of $\mathcal{G}$, we have that $\mathbb{E}_{\mathcal{U}}[b(x)] \leq \epsilon$ for all $b \in \mathcal{B}(\mathcal{G})$. Therefore, by Lemma 3.2 and Equation 6, we have

$$\mathbb{E}_{\mathcal{D}}\left[\max_{b\in\mathcal{B}(\mathcal{G})}\sum_{t=1}^{T}b(x_t)\right] \leq O\left(T\frac{\epsilon}{\sigma}\sqrt{\ln\left(\mathcal{N}_{[]}(\mathcal{H},\mathcal{U},\epsilon/4)\right)}\right)$$

Replacing this in Equation 5 we have that

$$\mathbb{E}[\text{REGRET}(\mathcal{A},\mathcal{D})] \in O\left(\sqrt{T\ln\left(\mathcal{N}_{[]}(\mathcal{H},\mathcal{U},\epsilon/4)\right)} + T\frac{\epsilon}{\sigma}\sqrt{\ln\left(\mathcal{N}_{[]}(\mathcal{H},\mathcal{U},\epsilon/4)\right)}\right)$$

Choosing $\epsilon = \sigma/\sqrt{T}$ proves the claim.

### C.4 Proof of Theorem 3.6

Consider the map $\psi : \mathcal{X} \to \mathbb{R}^m$ that embeds $\mathcal{G}$ in $m$ dimensions and let $\mathcal{H}$ be the class of halfspaces in $\mathbb{R}^m$. We want to bound the bracketing number of $\mathcal{G}$ by that of $\mathcal{H}$. Let $\mathcal{B}(\mathcal{H}) = \{[h_i, h^i]\}_i$ be an $\epsilon$-bracketing for $\mathcal{H}$ with respect to a measure $\mu$ that we will specify later. Consider the set of brackets $\mathcal{B}' = \{[h_i \circ \psi, h^i \circ \psi] \mid \text{for all } [h_i, h^i] \in \mathcal{B}(\mathcal{H})\}$. We first argue that $\mathcal{B}'$ is a bracketing for $\mathcal{G}$ with respect to $\nu$. To see this, note that any $g \in \mathcal{G}$ can be expressed as $g = h \circ \psi$ for some halfspace $h$. Considering the bracket $[h_i, h^i] \ni h$ in $\mathcal{B}(\mathcal{H})$. Note that $h_i \circ \psi \preceq h \circ \psi \preceq h^i \circ \psi$ and thus $g \in [h_i \circ \psi, h^i \circ \psi]$. We next argue that these are $\epsilon$-brackets under measure $\nu$. Let $\mu$ be the measure such that to sample $z \sim \mu$ we first sample $x \sim \nu$ and let $z = \psi(x)$. Note that

$$\Pr_{x\sim\nu}\left[h^i(\psi(x)) \neq h_i(\psi(x))\right] = \Pr_{z\sim\mu}\left[h^i(z) \neq h_i(z)\right] \leq \epsilon,$$

where the last transition is by the fact that $\mathcal{B}(\mathcal{H})$ is an $\epsilon$-bracketing for $\mathcal{H}$ with respect to $\mu$. This concludes that $\mathcal{B}'$ is an $\epsilon$-bracketing for $\mathcal{G}$ with respect to $\nu$. We complete the proof by using Theorem 3.4 to bound $|\mathcal{B}'| = |\mathcal{B}(\mathcal{H})| \leq (m/\epsilon)^{O(m)}$.

### C.5 Proof of Theorem 3.7

We first consider the case of $k = 2$ and then extend our argument to general $k$. Let $\epsilon' = \epsilon/k$ and let $\mathcal{B}(\mathcal{F}_1)$ and $\mathcal{B}(\mathcal{F}_2)$ be $\epsilon'$-bracketings for $\mathcal{F}_1$ and $\mathcal{F}_2$, respectively.

For $\mathcal{F}_1 \cdot \mathcal{F}_2$, construct $\mathcal{B} = \{[f_\ell \cap g_\ell, f^u \cap g^u] \mid \text{for all } [f_\ell, f^u] \in \mathcal{B}(\mathcal{F}_1) \text{ and } [g_\ell, g^u] \in \mathcal{B}(\mathcal{F}_2)\}$. First note for any $f_1 \in \mathcal{F}_1$ and $f_2 \in \mathcal{F}_2$, $f_1 \cap f_2$ is included in one of these brackets. In particular, for brackets $[f_\ell, f^u] \ni f_1$ and $[g_\ell, g^u] \ni f_2$, we have that $f_\ell \cap g_\ell \preceq f_1 \cap f_2 \preceq f^u \cap g^u$ and $[f_\ell \cap g_\ell, f^u \cap g^u] \in \mathcal{B}$. Furthermore,

$$\Pr_{x\sim\mu}\left[(f_\ell(x) \cap g_\ell(x)) \neq (f^u(x) \cap g^u(x))\right] \leq \Pr_{x\sim\mu}\left[(f_\ell(x) \cap g_\ell(x)) \neq (f_\ell(x) \cap g^u(x))\right]$$
$$+ \Pr_{x\sim\mu}\left[(f_\ell(x) \cap g^u(x)) \neq (f^u(x) \cap g^u(x))\right]$$
$$\leq 2\epsilon'.$$

Therefore, $\mathcal{B}$ is a $2\epsilon'$-bracketing for $\mathcal{F}_1 \cdot \mathcal{F}_2$ of size $\mathcal{N}_{[]}(\mathcal{F}_1,\mu,\epsilon') \cdot \mathcal{N}_{[]}(\mathcal{F}_2,\mu,\epsilon')$. Repeating this inductively and using $\epsilon' = \epsilon/k$, we get the claim for $k$ classes.

Similarly, for $\mathcal{F}_1 + \mathcal{F}_2$, construct $\mathcal{B} = \{[f_\ell \cup g_\ell, f^u \cup g^u] \mid \text{for all } [f_\ell, f^u] \in \mathcal{B}(\mathcal{F}_1) \text{ and } [g_\ell, g^u] \in \mathcal{B}(\mathcal{F}_2)\}$. First note for any $f_1 \in \mathcal{F}$ and $f_2 \in \mathcal{F}_1$ and their respective brackets $[f_\ell, f^u] \ni f_1$ and $[g_\ell, g^u] \ni f_2$, we have that $f_\ell \cup g_\ell \preceq f_1 \cup f_2 \preceq f^u \cup g^u$ and $[f_\ell \cup g_\ell, f^u \cup g^u] \in \mathcal{B}$. Furthermore,

$$\Pr_{x \sim \mu} \left[ (f_\ell(x) \cup g_\ell(x)) \neq (f^u(x) \cup g^u(x)) \right] \leq \Pr_{x \sim \mu} \left[ f_\ell(x) \neq f^u(x) \right] + \Pr_{x \sim \mu} \left[ g_\ell(x) \neq g^u(x) \right]$$

$$\leq 2\epsilon'.$$

Therefore, $\mathcal{B}$ is a $2\epsilon'$-bracketing for $\mathcal{F}_1 + \mathcal{F}_2$ of size $\mathcal{N}_{[]}(\mathcal{F}_1, \mu, \epsilon') \cdot \mathcal{N}_{[]}(\mathcal{F}_2, \mu, \epsilon')$. Repeating this inductively and using $\epsilon' = \epsilon/k$, we get the claim for $k$ classes.

As for the $\mathcal{G}$, the set of all symmetric differences, note that $f_1 \Delta f_2 = (f_1 \cup f_2) \setminus (f_1 \cap f_2) = (f_1 \cup f_2) \cap \overline{(f_1 \cap f_2)}$. Furthermore, for any class $\mathcal{F}$, the class $\overline{\mathcal{F}} = \{\overline{f} \mid \forall f \in \mathcal{F}\}$ has the same bracketing number as $\mathcal{F}$. Therefore, the bracketing number of $\mathcal{G}$ follows from using the bracketing number $\mathcal{F} + \mathcal{F}$, $\overline{\mathcal{F} + \mathcal{F}}$, and their intersection.

## C.6   Proof of Corollary 3.8

The set of polynomial threshold functions in $n$ variables and of degree $d$ is embeddable as halfspaces in $O(n^d)$ dimensions using the map

$$\phi(x_1, \ldots, x_n) = \left( \prod_{i \in S} x_i \right)_{S \in \{1, \ldots, n\}^{\leq d}},$$

which maps variables to all monomial of degree $d$. It can be seen that the number of monomials of degree at most $d$ in $n$ variables is given by $\binom{n+d+1}{d+1}$ which is approximately $O(n^d)$ when $d$ is small. Combining Theorem 3.6 and Theorem 3.4 completes the proof for polynomial threshold functions.

A $k$-polytope in $\mathbb{R}^n$ is an intersection of $k$-halfspaces in $\mathbb{R}^n$. Combining Theorem 3.7 and Theorem 3.4 completes the proof.

# D   More Details on Bracketing Number and Sign Rank

Though bracketing numbers are a fundamental concept in statistics, until recently their connection to VC theory was not well understood. Adams and Nobel [2010, 2012] show that for countable (can be generalized to classes that are well approximated by countable classes) classes with finite VC dimension the bracketing numbers with respect to any measure is finite (this establishes what is known as a universal Gilvenko–Cantelli theorem under ergodic sampling.)

**Theorem D.1** (Finite Bracketing Bounds for VC Classes)**.** *Let $\mathcal{C}$ be a countable class with finite VC dimension. Then, $\mathcal{N}_{[]}(\mathcal{C}, \mu, \epsilon) < \infty$.*

Though the above theorem proves that $\epsilon$-bracketing numbers are finite, their growth rate in $1/\epsilon$ can be arbitrarily large. See van Handel [2013] for some interesting examples of classes where the bracketing numbers grow arbitrarily fast.

Another combinatorial quantity that can help bound the regret in presence of adaptive smooth adversaries is *sign rank*.

**Definition D.2** (Sign Rank)**.** *Let $\mathcal{X}$ be an instance space and let $\mathcal{F}$ be a class. We can denote the class naturally as $\{-1, 1\}$-valued $\mathcal{X} \times \mathcal{F}$ matrix $M_{\mathcal{F}}$ where the entry corresponding to $(x, f)$ is $f(x)$. The sign rank of a class is the highest rank of a real matrix that agrees with a finite submatrix of $M_{\mathcal{F}}$ in sign. If this is unbounded, the class is said to have infinite sign rank.*

The sign rank of a class captures the dimension in which the class can be embedded as thresholds.

**Fact D.3** (Sign Rank Embedding, see e.g. Lokam [2009])**.** *The sign rank of a class corresponds to the smallest dimension $d$ that the class can be embedded as thresholds.*

Theorem 3.6 effectively says that classes with small sign rank have a slowly growing bracketing numbers and thus have low regret in the smoothed online learning setting. Thus, the complexity of smoothed online learning lies somewhere in between the sign rank and VC dimension. On the other

hand, it is known that even classes with small VC dimension can have arbitrarily large sign rank [Alon et al., 1987, Ben-David et al., 2003, Alon et al., 2016]. An intermediate question is whether classes with slow growing bracketing number also have good sign rank. It would be interesting to characterize the complexity of smoothed online learning in terms of either the sign rank or bracketing numbers.

# E   Query Answering

## E.1   Smooth MWEM Algorithm

---

**Algorithm 2:** Smooth Multiplicative Weights Exponential Mechanism

---

**Input:**   Universe $\mathcal{X}$ with $|\mathcal{X}| = N$, Data set $B$ with $n$ records, Query set $\mathcal{Q}$, Privacy parameters $\epsilon$ and $\delta$, Smoothness parameter $\sigma$.

Let $\mathcal{D}_0(x) = 1/N$ for all $x \in \mathcal{X}$.
*Cover Construction:* Compute $\mathcal{Q}' \subseteq \mathcal{Q}$ that is a $\gamma$-cover of $\mathcal{Q}$ with respect to the uniform distribution for $\gamma = \frac{\sigma}{2n}$.
**for** $i = 1 \ldots T$ **do**

  *Exponential Mechanism:* Sample $q_i \in \mathcal{Q}'$ according to the exponential mechanism with parameter $\epsilon/2T$ and score function

$$s_i(\mathcal{D}_B, q) = n\,|q(\mathcal{D}_{i-1}) - q(\mathcal{D}_B)|\,.$$

  *Laplace Mechanism:* Let $m_i = q_i(\mathcal{D}_B) + \frac{1}{n} Lap(2T/\epsilon)$ .
  *Multiplicative Update:* Update $\mathcal{D}_{i-1}$ using the rule

$$\mathcal{D}_i(x) \propto \mathcal{D}_{i-1}(x) \exp\left(\frac{q_i(x)(m_i - q_i(\mathcal{D}_{i-1}))}{2}\right).$$

**end**
Let $\overline{\mathcal{D}} = \frac{1}{T}\sum_{i=1}^T \mathcal{D}_i$.
**Output:** For each $q \in \mathcal{Q}$, answer with $v_q = q'(\overline{\mathcal{D}})$ where $q'$ is the closest function in $\mathcal{Q}'$ to $q$.

---

## E.2   Proof of Theorem 4.2

In this section we prove the following theorem.

**Theorem 4.2 (restated).** *For any $(\sigma, 0)$-smooth dataset $B$ of size $n$, a query class $\mathcal{Q}$ with VC dimension $d$, $T \in \mathbb{N}$ and $\epsilon > 0$, Smooth Multiplicative Weights Exponential Mechanism is $\epsilon$-differentially private and with probability at least $1 - 2T(\gamma/41)^{\mathrm{VCDim}(\mathcal{Q})}$, calculates values $v_q$ for all $q \in \mathcal{Q}$ such that*

$$\max_{q \in \mathcal{Q}}\{|v_q - q(\mathcal{D}_B)|\} \leq \frac{1}{n} + 2\sqrt{\frac{\log(1/\sigma)}{T}} + \frac{10Td\log(2n/\sigma)}{\epsilon n}.$$

Let us first provide a few useful lemmas.

**Lemma E.1** (Cover under Smoothness)**.** *Let $B$ be $(\sigma, 0)$-smooth data set. Let $\mathcal{Q}' \subseteq \mathcal{Q}$ be a $\gamma$-cover of $\mathcal{Q}$ under the uniform distribution. For a $q \in \mathcal{Q}$, let $q' \in \mathcal{Q}$ be such that $\mathrm{Pr}_{x \sim \mathcal{U}}[q(x) \neq q'(x)] \leq \gamma$. Then,*

$$|q(\mathcal{D}_B) - q'(\mathcal{D}_B)| \leq \frac{2\gamma}{\sigma}.$$

*Proof.* From the $(\sigma, 0)$-smoothness of $B$, we get

$$|q(\mathcal{D}_B) - q'(\mathcal{D}_B)| = \left|q(\overline{\mathcal{D}_B}) - q'(\overline{\mathcal{D}_B})\right|$$

$$\leq \sum_{x \in D} |(q(x) - q'(x))| \overline{\mathcal{D}_B}(x)$$

$$\leq \sum_{x \in \mathcal{X}} 2 \mathbb{I}(q(x) \neq q'(x)) \overline{\mathcal{D}_B}(x)$$

$$\leq \frac{2}{\sigma} \sum_{x \in \mathcal{X}} \mathbb{I}(q(x) \neq q'(x)) \mathcal{U}(x)$$

$$\leq \frac{2}{\sigma} \Pr_{x \sim \mathcal{U}} [q(x) \neq q'(x)]$$

$$\leq \frac{2\gamma}{\sigma}$$

as required. $\qquad\square$

Define the potential function $\Psi_i = \sum_{x \in \mathcal{X}} \overline{\mathcal{D}_B}(x) \log \left( \overline{\mathcal{D}_B}(x)/\mathcal{D}_i(x) \right)$, where $\overline{\mathcal{D}_B}$ is a corresponding $\sigma$-smooth distribution that matches the query answers for the $(\sigma, 0)$-smooth data set $B$. Here we make a few observations about the potential function.

**Fact E.2.** *For all $i \leq T$, we have $\Psi_i \geq 0$. Furthermore, $\Psi_0 \leq \log \frac{1}{\sigma}$. As a result, $\Psi_0 - \Psi_T \leq \log \frac{1}{\sigma}$.*

*Proof.* The first claim follows from the positivity of the KL divergence. For the second one, recall that from the $\sigma$-smoothness of $\mathcal{D}_B$ and the fact that $\mathcal{D}_1$ is the uniform distribution, we have $\mathcal{D}_B(x) \leq \sigma^{-1}\mathcal{D}_0(x)$ for all $x \in \mathcal{X}$.

$$\Psi_0 = \sum_{x \in \mathcal{X}} \overline{\mathcal{D}_B}(x) \log \frac{\overline{\mathcal{D}_B}(x)}{\mathcal{D}_0(x)} \leq \sum_{x \in \mathcal{X}} \overline{\mathcal{D}_B}(x) \log \frac{1}{\sigma} = \log \frac{1}{\sigma}$$

as required. $\qquad\square$

Below is a direct adaptation of a result of Hardt et al. [2012] for bounding the change in the potential functions.

**Lemma E.3** (Lemma A.4 in Hardt et al. [2012]).

$$\Psi_{i-1} - \Psi_i \geq \left( \frac{q_i(\mathcal{D}_{i-1}) - q_i(\overline{\mathcal{D}_B})}{2} \right)^2 - \left( \frac{m_i - q_i(\overline{\mathcal{D}_B})}{2} \right)^2.$$

**Lemma E.4** (Exponential and Laplace Mechanism guarantees). *With probability at least $1 - 2T/|\mathcal{Q}'|$, we have*

$$|q_i(\mathcal{D}_{i-1}) - q_i(\mathcal{D}_B)| \geq \max_{q' \in \mathcal{Q}'} \{q'(\mathcal{D}_i) - q'(\mathcal{D}_B)\} - \frac{8T \log |\mathcal{Q}'|}{\epsilon n}$$

*and*

$$|m_i - q_i(\mathcal{D}_B)| \leq \frac{2T \log |\mathcal{Q}'|}{\epsilon n}.$$

Here we recall again the error guarantees from Hardt et al. [2012].

**Theorem E.5** (Hardt et al. [2012]). *For any data set $B$ of size $n$, a finite query class $\mathcal{Q}$, $T \in \mathbb{N}$ and $\epsilon > 0$, MWEM is $\epsilon$-differentially private and with probability at least $1 - 2T/|\mathcal{Q}|$ produces a distribution $\overline{\mathcal{D}}$ over $\mathcal{X}$ such that*

$$\max_{q \in \mathcal{Q}} \left\{ |q(\overline{\mathcal{D}}) - q(\mathcal{D}_B)| \right\} \leq 2 \sqrt{\frac{\log |\mathcal{X}|}{T}} + \frac{10T \log |\mathcal{Q}|}{\epsilon n}.$$

*Proof of Theorem 4.2.* Our proof closely resembles that of Theorem E.5 from Hardt et al. [2012]. Note that since $B$ is $(\sigma, 0)$-smooth, we have a $\sigma$-smooth distribution $\overline{\mathcal{D}_B}$ with $\overline{\mathcal{D}_B}(x) \leq \frac{1}{\sigma N}$ such that for all $q \in \mathcal{Q}$, $q(\mathcal{D}_B) = q(\overline{\mathcal{D}_B})$. Furthermore, note that we chose a cover $\mathcal{Q}' \subseteq \mathcal{Q}$. Therefore, $q'(\mathcal{D}_B) = q'(\overline{\mathcal{D}_B})$ holds for all $q' \in \mathcal{Q}'$ as well.

Note that since $q'(\mathcal{D}_B) = q'(\overline{\mathcal{D}_B})$ for all $q' \in \mathcal{Q}'$, we can replace this in the above equation. For the sake of completeness, we sketch the rest of the proof. From Jensen's inequality, we have

$$\max_{q' \in \mathcal{Q}'} \left| q'(\overline{\mathcal{D}}) - q'(\mathcal{D}_B) \right| \leq \frac{1}{T} \sum_{i=1}^{T} \max_{q' \in \mathcal{Q}'} \left| q'(\mathcal{D}_i) - q'(\mathcal{D}_B) \right|. \tag{7}$$

From Lemma E.4 and Lemma E.3, we get that with probability at least $1 - 2T/|\mathcal{Q}'|$, we get

$$\Psi_{i-1} - \Psi_i \geq \left( \frac{\max_{q' \in \mathcal{Q}'} \{q'(\mathcal{D}_i) - q'(\mathcal{D}_B)\} - \frac{8T \log |\mathcal{Q}'|}{\epsilon n}}{2} \right)^2 - \left( \frac{T \log |\mathcal{Q}|}{\epsilon n} \right)^2.$$

Rearranging this and taking the average, we get

$$\frac{1}{T} \sum_{i=1}^{T} \max_{q' \in \mathcal{Q}'} \left| q'(\mathcal{D}_i) - q'(\mathcal{D}_B) \right| \leq \frac{1}{T} \sum_{i=1}^{T} \left[ \sqrt{4(\Psi_{i-1} - \Psi_i) + \frac{4T^2 \log^2 |\mathcal{Q}'|}{n^2 \epsilon^2}} + \frac{8T \log |\mathcal{Q}'|}{n\epsilon} \right].$$

Applying the concavity of the square root function i.e., $\frac{1}{T} \sum_{i=1}^{T} (x_i)^{1/2} \leq \left( \frac{1}{T} \sum_{i=1}^{T} x_i \right)^{1/2}$,

$$\frac{1}{T} \sum_{i=1}^{T} \max_{q \in \mathcal{Q}'} \left| q'(\mathcal{D}_i) - q'(\mathcal{D}_B) \right| \leq \sqrt{\sum_{i=1}^{T} \frac{4(\Psi_{i-1} - \Psi_i)}{T} + \frac{4T^2 \log^2 |\mathcal{Q}'|}{n^2 \epsilon^2}} + \frac{8T \log |\mathcal{Q}'|}{n\epsilon}$$

$$\leq \sqrt{\frac{4(\Psi_0 - \Psi_T)}{T} + \frac{4T^2 \log^2 |\mathcal{Q}'|}{n^2 \epsilon^2}} + \frac{8T \log |\mathcal{Q}'|}{n\epsilon}$$

$$\leq \sqrt{\frac{4 \log \left( \frac{1}{\sigma} \right)}{T} + \frac{4T^2 \log^2 |\mathcal{Q}'|}{n^2 \epsilon^2}} + \frac{8T \log |\mathcal{Q}'|}{n\epsilon}$$

$$\leq 2\sqrt{\frac{\log \left( \frac{1}{\sigma} \right)}{T}} + \frac{10T \log |\mathcal{Q}'|}{n\epsilon}.$$

The second inequality follows by summing the telescoping series. The third follows from Fact E.2. The last equation follows from the fact that $\sqrt{x+y} \leq \sqrt{x} + \sqrt{y}$ for all positive $x, y$. Using Equation 7 and the fact that $|\mathcal{Q}|' \leq (41/\gamma)^d$ we have

$$\max_{q' \in \mathcal{Q}'} \left| q'(\overline{\mathcal{D}}) - q'(\mathcal{D}_B) \right| \leq 2\sqrt{\frac{\log (1/\sigma)}{T}} + \frac{10Td \log (2n/\sigma)}{\epsilon n}.$$

Let $v_q = q'(\overline{\mathcal{D}})$ for $q' \in \mathcal{Q}'$ that is the closest hypothesis to $q$ with respect to the uniform distribution. Then

$$\left| q(\mathcal{D}_B) - v_q \right| = \left| q(\mathcal{D}_B) - q'(\mathcal{D}_B) + q'(\mathcal{D}_B) - q'(\overline{\mathcal{D}}) \right|$$

$$\leq \left| q(\mathcal{D}_B) - q'(\mathcal{D}_B) \right| + \left| q'(\mathcal{D}_B) - q'(\overline{\mathcal{D}}) \right|$$

$$\leq \frac{2\gamma}{\sigma} + 2\sqrt{\frac{\log 1/\sigma}{T}} + \frac{10Td \log (41/\gamma)}{\epsilon n}.$$

Setting $\gamma = \frac{\sigma}{4n}$, we get the desired result. $\qquad\square$

Setting $T = \epsilon^{2/3} n^{2/3} \log^{1/3}(1/\sigma) d^{-2/3} \log^{-2/3}(2n/\sigma)$, we get $(\epsilon, 0)$ differential privacy with

$$\max_{q \in \mathcal{Q}} \{ |v_q - q(\mathcal{D}_B)| \} \leq O\left( \sqrt[3]{\frac{d \log (1/\sigma) \log (2n/\sigma)}{n\epsilon}} \right).$$

Also, as noted in Hardt et al. [2012], one can use adaptive $k$-fold composition (see e.g. Dwork and Roth [2014]) to get $(\epsilon, \delta)$-differential privacy with

$$\max_{q \in \mathcal{Q}} \{ |v_q - q(\mathcal{D}_B)| \} \leq O\left( \sqrt{\frac{d}{\epsilon n} \log^{\frac{1}{2}} \left( \frac{1}{\sigma} \right) \log \left( \frac{n}{\sigma} \right) \log \left( \frac{1}{\delta} \right)} \right).$$

### E.3 Running Time of the Algorithm

The running time of the algorithm is similar to the running time of the MWEM algorithm of Hardt et al. [2012]. The main additional step is the construction of the cover $\mathcal{Q}'$. Similar to Appendix C.1 , this cover can be constructed in time $O\left(|\mathcal{Q}'|\right)$. The exponential mechanism requires $O\left(n|\mathcal{Q}|'\right)$ to evaluate all the queries on the cover and time $O\left(|\mathcal{Q}|'|\mathcal{X}|\right)$ to execute each iteration of the algorithm. Recall that $|\mathcal{Q}'| \leq (41n/\sigma)^d$, thus the running time is bounded by $O\left(n\left(41n/\sigma\right)^d + T\left(41n/\sigma\right)^d |\mathcal{X}|\right)$.

This runtime can also be improved using several theoretical tricks, e.g., $q(\mathcal{D}_i)$ can be approximated by taking random points from $\mathcal{D}_i$ in time that is independent of $\mathcal{X}$.

Note that the runtime of our algorithm improves upon the runtime of MWEM by using smaller query sets. As noted in Hardt et al. [2012], their algorithm is amenable to many optimizations and modifications that make it very fast and practical Hardt et al. [2012].

## F  Data Release

### F.1  Projected Smooth MWEM Algorithm

---

**Algorithm 3:** Projected Smooth Multiplicative Weight Exponential Mechanism

---

**Input:** Universe $\mathcal{X}$ with $|\mathcal{X}| = N$, Data set $B$ with $n$ records, Query set $\mathcal{Q}$, Privacy parameters $\epsilon$ and $\delta$, Smoothness parameter $\sigma$.

Let $\mathcal{D}_0\left(x\right) = 1/N$ for all $x \in \mathcal{X}$.
*Cover Construction:* Compute $\mathcal{Q}' \subseteq \mathcal{Q}$ that is a $\gamma$-cover of $\mathcal{Q}$ with respect to the uniform distribution for $\gamma = \frac{\sigma}{2n}$.
**for** $i = 1 \ldots T$ **do**

> *Exponential Mechanism:* Sample $q_i \in \mathcal{Q}'$ according to the exponential mechanism with parameter $\epsilon/2T$ and score function
>
> $$s_i(\mathcal{D}_B, q) = n\left|q\left(\mathcal{D}_{i-1}\right) - q(\mathcal{D}_B)\right|.$$
>
> *Laplace Mechanism:* Let $m_i = q_i\left(\mathcal{D}_B\right) + \frac{1}{n}Lap\left(2T/\epsilon\right)$.
> *Multiplicative Update:* Update $\mathcal{D}_{i-1}$ using the rule
>
> $$\tilde{\mathcal{D}}_i\left(x\right) \propto \mathcal{D}_{i-1}\left(x\right)\exp\left(\frac{q_i\left(x\right)\left(m_i - q_i(\mathcal{D}_{i-1})\right)}{2}\right).$$
>
> *KL Projection:* Project $\tilde{\mathcal{D}}_i$ onto the polytope $\mathcal{K} = \left\{\mathbf{z} : z_i \geq 0,\ \sum_{i=1}^{N} z_i = 1, z_i \leq \frac{1}{\sigma N}\right\}$ of smooth distributions:
>
> $$\mathcal{D}_i = \underset{\mathcal{D} \in \mathcal{K}}{\operatorname{argmin}}\, \mathrm{D_{KL}}(\mathcal{D}\|\tilde{\mathcal{D}}_i)$$

**end**
Let $\overline{\mathcal{D}} = \frac{1}{T}\sum_{i=1}^{T}\mathcal{D}_i$.
**Output:** Distribution $\overline{\mathcal{D}}$.

---

### F.2  Proof of Theorem 4.3

As before, let $\overline{\mathcal{D}_B}$ be a corresponding $\sigma$-smooth distribution that matches the query answers for the $(\sigma, 0)$-smooth data set $B$. Define $\Psi_i = \sum_{x \in \mathcal{X}} \overline{\mathcal{D}_B}(x)\log\left(\overline{\mathcal{D}_B}(x)/\mathcal{D}_i(x)\right)$ and $\tilde{\Psi}_i = \sum_{x \in \mathcal{X}} \overline{\mathcal{D}_B}(x)\log\left(\overline{\mathcal{D}_B}(x)/\tilde{\mathcal{D}}_i(x)\right)$ as the intermediate potential. From Lemma E.3, we know

$$\Psi_{i-1} - \tilde{\Psi}_i \geq \left(\frac{q_i\left(\mathcal{D}_{i-1}\right) - q_i(\mathcal{D}_B)}{2}\right)^2 - \left(\frac{m_i - q_i(\mathcal{D}_B)}{2}\right)^2.$$

Using the properties of relative entropy, we show the following claim.

**Claim F.1.** *For every $i \leq T$, we have $\tilde{\Psi}_i \geq \Psi_i$.*

*Proof.* The claim follows from the following fact about the KL divergence. Let

$$\mathcal{D}_i = \underset{\mathcal{D} \in \mathcal{K}}{\arg\min}\, \mathrm{D}_{\mathrm{KL}}(\mathcal{D}\|\tilde{\mathcal{D}}_i)$$

for some convex set $\mathcal{K}$. Then, for $\overline{\mathcal{D}_B} \in \mathcal{K}$,

$$\mathrm{D}_{\mathrm{KL}}(\overline{\mathcal{D}_B}\|\tilde{\mathcal{D}}_i) \geq \mathrm{D}_{\mathrm{KL}}\left(\overline{\mathcal{D}_B}\|\mathcal{D}_i\right) + \mathrm{D}_{\mathrm{KL}}\left(\mathcal{D}_i\|\tilde{\mathcal{D}}_i\right).$$

The claim follows by $\tilde{\Psi}_i = \mathrm{D}_{\mathrm{KL}}(\mathcal{D}_B\|\tilde{\mathcal{D}}_i)$, $\Psi_i = \mathrm{D}_{\mathrm{KL}}\left(\mathcal{D}_B\|\mathcal{D}_i\right)$ and $\mathrm{D}_{\mathrm{KL}}\left(\mathcal{D}_i\|\tilde{\mathcal{D}}_i\right) \geq 0$. $\qquad\square$

Together this gives

$$\Psi_{i-1} - \Psi_i \geq \left(\frac{q_i\left(\mathcal{D}_{i-1}\right) - q_i(\mathcal{D}_B)}{2}\right)^2 - \left(\frac{m_i - q_i(\mathcal{D}_B)}{2}\right)^2.$$

The remainder of the analysis follows that of Theorem 4.2. Note that we have $\overline{\mathcal{D}}$ is $\sigma$-smooth since each $\mathcal{D}_i \in \mathcal{K}$ and $\mathcal{K}$ is a convex set. By Lemma E.1, we have $\left|q'\left(\overline{\mathcal{D}}\right) - q\left(\overline{\mathcal{D}}\right)\right| \leq {2\gamma}/{\sigma}$. Thus,

$$\begin{aligned}
\left|q\left(\mathcal{D}_B\right) - q\left(\overline{\mathcal{D}}\right)\right| &= \left|q\left(\mathcal{D}_B\right) - q'\left(\mathcal{D}_B\right) + q'\left(\mathcal{D}_B\right) - q'\left(\overline{\mathcal{D}}\right) + q'\left(\overline{\mathcal{D}}\right) - q\left(\overline{\mathcal{D}}\right)\right| \\
&\leq \left|q\left(\mathcal{D}_B\right) - q'\left(\mathcal{D}_B\right)\right| + \left|q'\left(\mathcal{D}_B\right) - q'\left(\overline{\mathcal{D}}\right)\right| + \left|q'\left(\overline{\mathcal{D}}\right) - q\left(\overline{\mathcal{D}}\right)\right| \\
&\leq \frac{4\gamma}{\sigma} + 2\sqrt{\frac{\log {1}/{\sigma}}{T}} + \frac{10Td\log\left({41}/{\gamma}\right)}{\epsilon n}.
\end{aligned}$$

Setting $\gamma = {\sigma}/{4n}$, we get

$$\left|q\left(\mathcal{D}_B\right) - q\left(\overline{\mathcal{D}}\right)\right| = \frac{1}{n} + 2\sqrt{\frac{\log\left({1}/{\sigma}\right)}{T}} + \frac{10Td\log\left({4n}/{\sigma}\right)}{\epsilon n}.$$

### F.3 Running Time of Projected Smooth Multiplicative Weights Exponential Mechanism

The running time is similar to the running time Smooth Multiplicative Weights Exponential Mechanism, with the additional projection step in each step. Note that the projection in each step is a convex program and can be solved in time $\mathrm{poly}\left(|\mathcal{X}|\right)$. This gives us a total running time of $O\left(n\left({41n}/{\sigma}\right)^d + T\left({41n}/{\sigma}\right)^d|\mathcal{X}| + T\mathrm{poly}(|\mathcal{X}|)\right).$

In addition to the improvements discussed in the previous sections, the projection step can be performed faster by taking an approximate Bregman projection as considered by Barak et al. [2009]. Incorporating this into our algorithm would lead to significant speed ups.

# G Smooth Data Release using SmallDB Algorithm

In this section,, we look at a different algorithm to get differential privacy when dealing with $(\sigma, \chi)$-smooth data sets. Our algorithm displayed below uses several pieces that have been introduced by Blum et al. [2008] and Hardt and Rothblum [2010].

---

**Algorithm 4:** Subsampled Net Mechanism

---

**Input:** Database $B$ of size $n$, Query set $\mathcal{Q}$, Privacy parameter $\epsilon$, Subsampling parameter $M$,
 Accuracy parameter $\gamma$.

Sample (with replacement) a subset $V$ of size $M$ from $\mathcal{X}$.
Sample $B'$ from amongst all data sets supported on $V$ of size

$$O\left(\frac{d}{\gamma^2}\right)$$

with probability proportional to

$$\exp\left(-\frac{\epsilon \cdot n \cdot s\left(\mathcal{D}_{B'}, \mathcal{D}_B\right)}{2}\right)$$

where $s\left(\mathcal{D}_{B'}, \mathcal{D}_B\right) = \max_{q \in \mathcal{Q}} |q(\mathcal{D}_B) - q(\mathcal{D}_{B'})|$.

**Output:** Database $B'$

---

First, we analyze the privacy of this algorithm.

**Theorem G.1.** *The Subsampled Net Mechanism is $(\epsilon, 0)$ differentially private.*

*Proof.* The privacy claim follows from the privacy of the exponential mechanism. □

Next we bound the error of this mechanism. Let us recall the standard uniform convergence bound.

**Fact G.2** (Uniform Convergence for VC Classes, see e.g. Shalev-Shwartz and Ben-David [2014])**.**
*Let $\mathcal{X}$ be the domain, $\mathcal{Q}$ be a class of queries over $\mathcal{X}$ with VC dimension $d$ and let $\mathcal{D}$ be a distribution. Let $\mathcal{D}'$ be a distribution gotten by sampling $O\left((\log(2/\eta) + d)/\gamma^2\right)$ items iid from $\mathcal{D}$ and normalizing the frequencies. Then, with probability $1 - \eta$, for all $q \in \mathcal{Q}$, $|q(\mathcal{D}') - q(\mathcal{D})| \leq \gamma$.*

In the following, we use the above fact to show that a randomly sampled subset of the universe approximates a $(\sigma, \chi)$-smooth database. The proof largely follows the domain reduction lemma of Hardt and Rothblum [2010] that achieve a similar bond by with a dependence on $\log(|\mathcal{Q}|)$. We include this proof for completeness.

**Lemma G.3.** *Let $\mathcal{X}$ be a data universe and $\mathcal{Q}$ a collection of queries over $\mathcal{X}$ with VC dimension $d$ and $\mathcal{D}$ be $(\sigma, \chi)$-smooth with respect to $\mathcal{Q}$. Let $V \subset \mathcal{X}$ of size $M$ be sampled from $\mathcal{X}$ at random with replacement with*

$$M = O\left(\frac{\log(1/\eta) + d}{\sigma\gamma^2}\right).$$

*Then, with probability $1 - \eta$, there exists a $\mathcal{D}'$ on $V$ such that for all $q \in \mathcal{Q}$*

$$|q(\mathcal{D}) - q(\mathcal{D}')| \leq \chi + \gamma.$$

*Proof.* Let $\mathcal{D}_1$ be $\sigma$-smooth distribution that witnesses the $(\sigma, \chi)$-smoothness of $\mathcal{D}$. If we could sample from $\mathcal{D}_1$, we would be done from Fact G.2. But we want to get a subset that is oblivious to the distribution $\mathcal{D}$. To achieve this, we use the smoothness of $\mathcal{D}_1$.

The idea is to sample from $\mathcal{D}_1$ using rejection sampling. Since $\mathcal{D}_1$ is $\sigma$-smooth, the following procedure produces samples from $\mathcal{D}_1$: sample from the uniform distribution and accept sample $u$ with probability $\sigma N \mathcal{D}_1(u)$. Note that accepted samples are distributed according to $\mathcal{D}_1$. We repeat this process until $O\left((\log(2/\eta) + d)/\gamma^2\right)$ samples are accepted. Since the accepted samples are distributed according to $\mathcal{D}_1$, from Fact G.2, there is a distribution $\mathcal{D}_2$ supported on the accepted samples such that with probability at least $1 - \eta/2$ for all $q \in \mathcal{Q}$,

$$|q(\mathcal{D}_2) - q(\mathcal{D})| \leq \chi + \gamma.$$

Let $S_1$ be the coordinates corresponding to the accepted samples and $S_2$ be the coordinates corresponding to the rejected ones. The key observation is that $S = S_1 \cup S_2$ is subset generated by sampling from the uniform distribution and has a distribution supported on it that approximates $\mathcal{D}$. So, it suffices to bound the size of $S$. The probability that a given sample gets accepted is

$$\sum_{x \in \mathcal{X}} \frac{\mathcal{D}_1(x) N \sigma}{N} = \sigma.$$

Thus the expected number of samples needed in the rejection sampling procedure is $M = O\left(\frac{\log(2/\eta) + d}{\sigma \gamma^2}\right)$. Using a Chernoff bound, we can bound the probability that this is greater than its mean by a factor of $4$ by

$$e^{-M} \leq \frac{\eta}{2}$$

where we used that fact that $M \geq \log(2/\eta)$. $\qquad\square$

We are finally ready to prove our theorem.

**Theorem G.4.** *For any data set $B$ that is $(\sigma, \chi)$-smooth with respect to a set of queries $\mathcal{Q}$ of VC dimension $d$, the output $\mathcal{D}''$ of the Subsampled Net Mechanism satisfies that with probability $1 - \eta$, for all $q \in \mathcal{Q}$*

$$|q(\mathcal{D}_B) - q(\mathcal{D}'')| \leq \chi + \tilde{O}\left(\sqrt[3]{\frac{d \log(1/\sigma) + \log(1/\eta)}{\epsilon n}}\right)$$

*Proof.* Consider a subset $V$ sampled with size $M = O\left(\frac{\log(1/\eta_1) + d}{\sigma \gamma^2}\right)$ where $\eta_1$ and $\gamma$ are parameters we will set later. From Lemma G.3, with probability $1 - \eta_1$ we have that there exists a distribution $\mathcal{D}'$ supported on $V$ such that for all $q \in \mathcal{Q}$

$$|q(\mathcal{D}') - q(\mathcal{D}_B)| \leq \chi + \gamma.$$

Let us work conditioned on this event. Let $A$ denote the set of all data sets supported on $V$ and let $C$ denote all data sets supported on $V$ with size $O\left(d\gamma^{-2}\right)$. From Fact G.2, for any distribution $\mathcal{D}_1$ supported on $V$, there is a data set in $C$ whose distribution $\mathcal{D}_2$ satisfies

$$|q(\mathcal{D}_1) - q(\mathcal{D}_2)| \leq \gamma.$$

We recall the guarantees of the exponential mechanism (see e.g. Dwork and Roth [2014]): Let $B''$ be the data base output by the exponential mechanism. Then,

$$\Pr\left[s(\mathcal{D}_{B''}, \mathcal{D}_B) \geq \min_{B_1 \in C} s(\mathcal{D}_{B_1}, \mathcal{D}_B) - \frac{2}{\epsilon n}(\log|C| + t)\right] \leq e^{-t},$$

where $s(\mathcal{D}_B, \mathcal{D}_{B'}) = \max_{q \in \mathcal{Q}} |q(\mathcal{D}_B) - q(\mathcal{D}_{B'})|$. Note that $\log|C| \leq M^{O(d\gamma^{-2})}$. Thus, with probability $1 - \eta_2$,

$$s(\mathcal{D}_{B''}, \mathcal{D}_B) \geq \min_{B_1 \in C} s(\mathcal{D}_{B_1}, \mathcal{D}_B) - \gamma$$

for

$$\gamma \geq \frac{4}{\epsilon n} \log \frac{M^{O(d\gamma^{-2})}}{\eta_2}.$$

Since, $\min_{B_1 \in C} s(\mathcal{D}_{B_1}, \mathcal{D}_B) \leq \chi + 2\gamma$, setting $\eta_1 = \eta_2 = \eta/2$ and solving for $\gamma$, we get

$$\gamma = \tilde{O}\left(\sqrt[3]{\frac{d \log(1/\sigma) + \log(1/\eta)}{\epsilon n}}\right)$$

as required. $\qquad\square$

### G.1 Running Time of Subsampled Net Mechanism

The running time of the algorithm involves first sampling $M$ elements uniformly from the domain which takes time $O(M \log|\mathcal{X}|)$. Each query needs to be evaluated on the data set $B$ which takes time $n|\mathcal{Q}|$. Evaluating and sampling from all data bases as required by the exponential mechanism naively takes time $M^{O(d\gamma^{-2})}$. As discussed earlier, this can be sped up using sampling for approximation.

[Supplementary Material 2 · appendix.pdf]

# A   Additional Related Work

Analogous models of smoothed online learning have been explored in prior work. Rakhlin et al. [2011] consider online learning when the adversary is constrained in several ways and work with a notion of sequential Rademacher complexity for analyzing the regret. In particular, they study a related notion of smoothed adversary and show that one can learn thresholds with regret of $O(\sqrt{T})$ in presence of smoothed adversaries. Gupta and Roughgarden [2017] consider smoothed online learning in the context online algorithm design. They show that while optimizing parameterized greedy heuristics for Maximum Weight Independent Set imposes linear regret in the worst-case, in presence of smoothing this problem can be learned with sublinear regret (as long they allow per-step runtime that grows with $T$). Cohen-Addad and Kanade [2017] consider the same problem with an emphasis on the per-step runtime being logarithmic in $T$. They show that piecewise constant functions over the interval $[0, 1]$ can be learned efficiently within regret of $O(\sqrt{T})$ against a *non-adaptive* smooth adversary. Our work differs from these by upper bounding the regret using a combinatorial dimension of the hypothesis class and demonstrating techniques that generalize to large class of problems in presence of *adaptive* adversaries.

In another related work, Balcan et al. [2018] introduce a notion of dispersion in online optimization (where the learner picks an instance and the adversary picks a function) that is a constraint on the number of discontinuities in the adversarial sequence of functions. They show that online optimization can be done efficiently under certain assumptions. Moreover, they show that sequences generated by *non-adaptive* smooth adversaries in one dimension satisfy dispersion. In comparison, our main results in online learning consider the more powerful adaptive adversaries.

Smoothed analysis is also used in a number of other online settings. In the setting of linear contextual bandits, Kannan et al. [2018] use smoothed analysis to show that the greedy algorithm achieves sublinear regret even though in the worst case it can have linear regret. Raghavan et al. [2018] work in a Bayesian version of the same setting and achieve improved regret bounds for the greedy algorithm. Since several algorithms are known to have sublinear regret in the linear contextual bandit setting even in the worst-case, the main contribution of these papers is to show that the simple and practical greedy algorithm has much better regret guarantees than in the worst-case. In comparison, we work with a setting where no algorithm can achieve sublinear regret in the worst-case.

Smoothed analysis has also been considered in the context of differential privacy. Hardt and Rothblum [2010] consider differential privacy in the interactive setting, where the queries arrive online. They analyze a multiplicative weights based algorithm whose running time and error they show can be vastly improved in the presence of smoothness. Some of our techniques for query answering and data release are inspired by that line of work. Balcan et al. [2018] also differential privacy in presence of dispersion and analyze the gaurantees of the exponential mechanism.

Generally, our work is also related to a line of work on online learning in presence of additional assumptions resembling properties exhibited by real life data. Rakhlin and Sridharan [2013] consider settings where additional information in terms of an estimator for future instances is available to the learner. They achieve regret bounds that are in terms of the path length of these estimators and can beat $\Omega(\sqrt{T})$ if the estimators are accurate. Dekel et al. [2017] also considers the importance of incorporating side information in the online learning framework and show that regrets of $O(\log(T))$ in online linear optimization maybe possible when the learner knows a vector that is weakly correlated with the future instances.

More broadly, our work is among a growing line of work on beyond the worst-case analysis of algorithms [Roughgarden, 2020] that considers the design and analysis of algorithms on instances that satisfy properties demonstrated by real-world applications. Examples of this in theoretical machine learning mostly include improved runtime and approximation guarantees of numerous supervised (e.g., [Kalai et al., 2009, Kalai and Teng, 2008, Awasthi et al., 2016, Diakonikolas et al., 2019]), and unsupervised settings (e.g., [Bilu and Linial, 2012, Balcan et al., 2020, 2013, Arora et al., 2012, Bhaskara et al., 2019, Vijayaraghavan et al., 2017, Makarychev et al., 2014, Ostrovsky et al., 2013, Hardt and Roth, 2013]).

# B   Lack of Uniform Convergence with Adaptive Adversaries

The following example for showing lack of uniform convergence over adaptive sequences is due to Haghtalab [2018] and is included here for completeness.

Let $\mathcal{X} = [0, 1]$ and $\mathcal{G} = \{g_b(x) = \mathbb{I}(x \geq b) \mid \forall b \in [0, 1]\}$ be the set of one-dimensional thresholds. Let the distribution of the noise $\eta_i$ be the uniform distribution on $(-1/4, 1/4)$. Let $x_1 = 1/2$ and $x_2 = x_3 = \cdots = x_T = 1/4$ if $\eta_1 \leq 0$ while $x_2 = x_3 = \cdots = x_T = 3/4$ otherwise. In this case, we do not achieve concentration for any value of $T$, as

$$\frac{1}{T} \sum_{t=1}^{T} g_{0.5}(x_t + \eta_t) = \begin{cases} 0 & \text{w.p. } 1/2 \\ 1 & \text{w.p. } 1/2 \end{cases} \quad and \quad \mathbb{E}\left[\frac{1}{T} \sum_{t=1}^{T} g_{0.5}(x_t + \eta_t)\right] = \frac{1}{2}.$$

# C   Proofs from Section 3

## C.1   Algorithm and its Running Time

While our main focus is to provide sublinear regret bounds for smoothed online learning our analysis also provides an algorithmic solution describe below.

---
**Algorithm 1:** Smooth Online Learning

**Input:** Instance Space $\mathcal{X}$, Hypothesis Class $\mathcal{H}$, Smoothness parmeter $\sigma$, Time horizon $T$

*Cover Construction:* Compute $\mathcal{H}' \subseteq \mathcal{H}$ that is a $\gamma$-cover of $\mathcal{H}$ with respect to the uniform distribution on $\mathcal{X}$ for $\gamma = \frac{\sigma}{4\sqrt{T}}$.

**for** $t = 1 \ldots T$ **do**
  Use a standard online learning algorithm, such as Hedge, on $\mathcal{H}'$ to pick an $h_t$, where the history of the play is $\{s_\tau\}_{\tau < t}$ and $\{h_\tau\}_{\tau < t}$
  Receive $s_t = (x_t, y_t)$ and suffer loss $\text{err}_{s_t}(h_t)$.
**end**

---

The running time of the algorithm comprises of the initial construction of $\mathcal{H}'$ and then running a standard online learning algorithm on $\mathcal{H}'$.

Standard online learning algorithms such as Hedge and FTPL take time polynomial in the size of the cover since in standard implementations they maintain a state corresponding to each hypothesis in $\mathcal{H}'$. In our setting, the size of the cover is $(41\sqrt{T}/\sigma)^d$.

The time required to construct a cover depends on the access we have to the class. One method is to randomly sample a set $S$ with $m = O(\text{VCDim}(\mathcal{H})\, T/\sigma^2)$ points from the domain uniformly and construct all possible labelings on this set induced by the class. The number of labellings of $S$ is bounded by $O(m^{\text{VCDim}(\mathcal{H})})$ by the Sauer–Shelah lemma. The cover is constructed by then finding functions in the class $\mathcal{H}$ that are consistent with each of these labellings. This requires us to be able to find an element in the class consistent with a given labeling, which can be done by a "consistency" oracle. Naively, the above makes $2^m$ calls to the consistency oracle, one for each possible labeling of $S$.

The above analysis and runtime can be improved in several ways. First, $\mathcal{H}'$ can be constructed in time $O(m^{\text{VCDim}(\mathcal{H})})$ rather than $2^m$. This can be done by constructing the cover in a hierarchical fashion, where the root includes the unlabeled set $S$ and at every level one additional instance in $S$ is labeled by $+1$ or $-1$. At each node, the consistency oracle will return a function $h \in \mathcal{H}$ that is consistent with the labels so far or state that none exists. Nodes for which no consistent hypothesis so far exists are pruned and will not expand in the next level. Since the total number of leaves is the number of ways in which $S$ can be labeled by $\mathcal{H}$, i.e., $O(m^d)$, the number of calls to the consistency oracle is $O(m^d)$ as well. The runtime of standard online learning algorithms can also be improved significantly when an empirical risk minimization oracle is available to the learner, in which case a runtime of $O(\sqrt{|\mathcal{H}'|})$ for general classes [Hazan and Koren, 2016] or even $\text{polylog}(|\mathcal{H}'|)$ for structured classes [Dudík et al., 2017] is possible.

## C.2  Proof of Lemma 3.2

At a high level, note that any $f \in \mathcal{F}$ has measure at most $\epsilon/\sigma$ on any (even adaptively chosen) $\sigma$-smooth distribution. Therefore, for any fixed $f$, $\mathbb{E}_{\mathscr{D}}[\sum_{t=1}^{T} f(x_t)] \leq T\epsilon/\sigma$. To achieve this bound over all $f \in \mathcal{F}$, we take a union bound over all such functions.

More formally, for any $s$

$$\exp\left(s \mathop{\mathbb{E}}_{\mathscr{D}}\left[\max_{f \in \mathcal{F}} \sum_{t=1}^{T} f(x_t)\right]\right) \leq \mathop{\mathbb{E}}_{\mathscr{D}}\left[\exp\left(s \max_{f \in \mathcal{F}} \sum_{t=1}^{T} f(x_t)\right)\right] \qquad \text{(Jensen's inequqlity)}$$

$$\leq \mathop{\mathbb{E}}_{\mathscr{D}}\left[\max_{f \in \mathcal{F}} \exp\left(s \sum_{t=}^{T} f(x_t)\right)\right] \qquad \text{(Monotonicity of } \exp\text{)}$$

$$\leq \sum_{f \in \mathcal{F}} \mathop{\mathbb{E}}_{\mathscr{D}}\left[\exp\left(s \sum_{t=1}^{T} f(x_t)\right)\right]. \tag{3}$$

Consider a fixed $f \in \mathcal{F}$. Note that even when the choice of a $\sigma$-smoothed distribution $\mathcal{D}$ depends on earlier realizations of $x_1, \dots, x_{i-1}$, $\Pr_{x_i \sim \mathcal{D}}[f(x_i)] \leq \frac{\epsilon}{\sigma}$. Therefore, $\sum_{t=1}^{T} f(x_t)$ for $\mathbf{x} \sim \mathscr{D}$ is stochastically dominated by that of a binomial distribution $Bin(T, \epsilon/\sigma)$. Note that $\exp(\cdot)$ is a monotonically increasing functions and let $p = \epsilon/\sigma$. We have

$$\mathop{\mathbb{E}}_{\mathscr{D}}\left[\exp\left(s \sum_{t=1}^{T} f(x_t)\right)\right] \leq \sum_{v=0}^{T} \exp(sv)\binom{T}{v} p^v (1-p)^{T-v} = \left(p(\exp(s) - 1) + 1\right)^T. \tag{4}$$

Combining Equations (3) and (4) and noting that $\ln(1+x) \leq x$, we have

$$\mathop{\mathbb{E}}_{\mathscr{D}}\left[\max_{f \in \mathcal{F}} \sum_{t=1}^{T} f(x_t)\right] \leq \frac{\ln(|\mathcal{F}|) + Tp\left(\exp(s) - 1\right)}{s}.$$

Let $s = \sqrt{\ln(|\mathcal{F}|)/Tp}$. Note that because $s \in (0, 1)$, we have $\exp(s) \leq 1 + 2s$. Hence, by replacing $s$ in the above inequality we have

$$\mathop{\mathbb{E}}_{\mathscr{D}}\left[\max_{f \in \mathcal{F}} \sum_{t=1}^{T} f(x_t)\right] \in O\left(Tp\sqrt{\ln(|\mathcal{F}|)}\right).$$

## C.3  Proof of Theorem 3.3

Consider any hypothesis class $\mathcal{H}'$ and an algorithm that is no-regret with respect to any adaptive adversary on hypotheses in $\mathcal{H}'$. It is not hard to see that

$$\mathbb{E}[\text{REGRET}(\mathcal{A}, \mathscr{D})] = \mathop{\mathbb{E}}_{\mathbf{s} \sim \mathscr{D}}\left[\sum_{t=1}^{T} \text{err}_{s_t}(h_t) - \min_{h \in \mathcal{H}} \text{err}_{s_t}(h_t)\right]$$

$$\leq \mathop{\mathbb{E}}_{\mathbf{s} \sim \mathscr{D}}\left[\sum_{t=1}^{T} \text{err}_{s_t}(h_t) - \min_{h \in \mathcal{H}'} \sum_{t=1}^{T} \text{err}_{s_t}(h)\right] + \mathop{\mathbb{E}}_{\mathbf{s} \sim \mathscr{D}}\left[\min_{h' \in \mathcal{H}'} \sum_{t=1}^{T} \text{err}_{s_t}(h') - \min \sum_{t=1}^{T} \text{err}_{s_t}(h)\right]$$

$$\leq O\left(\sqrt{T \ln(|\mathcal{H}'|)}\right) + \mathop{\mathbb{E}}_{\mathscr{D}}\left[\max_{h \in \mathcal{H}} \min_{h' \in \mathcal{H}'} \sum_{t=1}^{T} \mathbb{1}\left(h(x_t) \neq h'(x_t)\right)\right]. \tag{5}$$

Therefore, it is sufficient to choose an $\mathcal{H}'$ of moderate size such that every function $h \in \mathcal{H}$ has a proxy $h' \in \mathcal{H}'$ even when these functions are evaluated on instances drawn from a *non-iid and adaptive sequence of smooth distributions*. We next describe the choice of $\mathcal{H}'$.

Let $\mathcal{H}'$ be a $\frac{\epsilon}{2}$-net of $\mathcal{H}$ with respect to the uniform distribution $\mathcal{U}$, for an $\epsilon$ that we will determine later. Note that any $\epsilon$-bracket with respect to $\mathcal{U}$ is also an $\epsilon$-net, so $|\mathcal{H}'| \leq \mathcal{N}_{[]}(\mathcal{H}, \mathcal{U}, \epsilon/2)$.[3] Let $\mathcal{G}$ be the set of symmetric differences between $h \in \mathcal{H}$ and its closest proxy $h' \in \mathcal{H}'$, that is,

$$\mathcal{G} = \{g_{h,h'}(x) = \mathbb{1}(h(x) \neq h'(x)) \mid \forall h \in \mathcal{H} \text{ and } h' \in \mathcal{H}', \text{ s.t. } \mathop{\mathbb{E}}_{\mathcal{U}}[g_{h,h'}(x)] \leq \epsilon/2\}.$$

Note that because $\mathcal{G}$ is a subset of all the symmetric differences of two functions in $\mathcal{H}$, by Theorem 3.7 its bracketing number is bounded as follows.

$$\mathcal{N}_{[]}(\mathcal{G},\mathcal{U},\epsilon/2) \leq \left(\mathcal{N}_{[]}(\mathcal{H},\mathcal{U},\epsilon/4)\right)^4. \tag{6}$$

Let $\mathcal{B}(\mathcal{G})$ be the set of upper $\epsilon/2$-brackets of $\mathcal{G}$ with respect to $\mathcal{U}$, i.e., for all $g \in G$, there is $b \in \mathcal{B}(\mathcal{G})$ such that for all $x \in \mathcal{X}$, $g(x) \leq b(x)$ and $\mathbb{E}_{\mathcal{U}}[b(x) - g(x)] \leq \epsilon/2$. Note that

$$\mathbb{E}_{\mathscr{D}}\left[\max_{h\in\mathcal{H}}\min_{h'\in\mathcal{H}'}\sum_{t=1}^{T} 1\left(h(x_t) \neq h'(x_t)\right)\right] = \mathbb{E}_{\mathscr{D}}\left[\max_{g\in\mathcal{G}}\sum_{t=1}^{T} g(x_t)\right] \leq \mathbb{E}_{\mathscr{D}}\left[\max_{b\in\mathcal{B}(\mathcal{G})}\sum_{t=1}^{T} b(x_t)\right],$$

where the last transition is by the fact that $\mathcal{B}(\mathcal{G})$ includes all upper brackets of $\mathcal{G}$.

We now note that $\mathcal{B}(\mathcal{G})$ meets the conditions Lemma 3.2, namely because all $g \in \mathcal{G}$ have measure at most $\epsilon/2$ over $\mathcal{U}$ and $\mathcal{B}(\mathcal{G})$ is the set of $\epsilon/2$-upper brackets of $\mathcal{G}$, we have that $\mathbb{E}_{\mathcal{U}}[b(x)] \leq \epsilon$ for all $b \in \mathcal{B}(\mathcal{G})$. Therefore, by Lemma 3.2 and Equation 6, we have

$$\mathbb{E}_{\mathscr{D}}\left[\max_{b\in\mathcal{B}(\mathcal{G})}\sum_{t=1}^{T} b(x_t)\right] \leq O\left(T\frac{\epsilon}{\sigma}\sqrt{\ln\left(\mathcal{N}_{[]}(\mathcal{H},\mathcal{U},\epsilon/4)\right)}\right)$$

Replacing this in Equation 5 we have that

$$\mathbb{E}[\text{REGRET}(\mathcal{A},\mathscr{D})] \in O\left(\sqrt{T\ln\left(\mathcal{N}_{[]}(\mathcal{H},\mathcal{U},\epsilon/4)\right)} + T\frac{\epsilon}{\sigma}\sqrt{\ln\left(\mathcal{N}_{[]}(\mathcal{H},\mathcal{U},\epsilon/4)\right)}\right)$$

Choosing $\epsilon = \sigma/\sqrt{T}$ proves the claim.

## C.4   Proof of Theorem 3.6

Consider the map $\psi : \mathcal{X} \to \mathbb{R}^m$ that embeds $\mathcal{G}$ in $m$ dimensions and let $\mathcal{H}$ be the class of halfspaces in $\mathbb{R}^m$. We want to bound the bracketing number of $\mathcal{G}$ by that of $\mathcal{H}$. Let $\mathcal{B}(\mathcal{H}) = \{[h_i, h^i]\}_i$ be an $\epsilon$-bracketing for $\mathcal{H}$ with respect to a measure $\mu$ that we will specify later. Consider the set of brackets $\mathcal{B}' = \{[h_i \circ \psi, h^i \circ \psi] \mid \text{for all } [h_i, h^i] \in \mathcal{B}(\mathcal{H})\}$. We first argue that $\mathcal{B}'$ is a bracketing for $\mathcal{G}$ with respect to $\nu$. To see this, note that any $g \in \mathcal{G}$ can be expressed as $g = h \circ \psi$ for some halfspace $h$. Considering the bracket $[h_i, h^i] \ni h$ in $\mathcal{B}(\mathcal{H})$. Note that $h_i \circ \psi \preceq h \circ \psi \preceq h^i \circ \psi$ and thus $g \in [h_i \circ \psi, h^i \circ \psi]$. We next argue that these are $\epsilon$-brackets under measure $\nu$. Let $\mu$ be the measure such that to sample $z \sim \mu$ we first sample $x \sim \nu$ and let $z = \psi(x)$. Note that

$$\Pr_{x\sim\nu}\left[h^i(\psi(x)) \neq h_i(\psi(x))\right] = \Pr_{z\sim\mu}\left[h^i(z) \neq h_i(z)\right] \leq \epsilon,$$

where the last transition is by the fact that $\mathcal{B}(\mathcal{H})$ is an $\epsilon$-bracketing for $\mathcal{H}$ with respect to $\mu$. This concludes that $\mathcal{B}'$ is an $\epsilon$-bracketing for $\mathcal{G}$ with respect to $\nu$. We complete the proof by using Theorem 3.4 to bound $|\mathcal{B}'| = |\mathcal{B}(\mathcal{H})| \leq (m/\epsilon)^{O(m)}$.

## C.5   Proof of Theorem 3.7

We first consider the case of $k = 2$ and then extend our argument to general $k$. Let $\epsilon' = \epsilon/k$ and let $\mathcal{B}(\mathcal{F}_1)$ and $\mathcal{B}(\mathcal{F}_2)$ be $\epsilon'$-bracketings for $\mathcal{F}_1$ and $\mathcal{F}_2$, respectively.

For $\mathcal{F}_1 \cdot \mathcal{F}_2$, construct $\mathcal{B} = \{[f_\ell \cap g_\ell, f^u \cap g^u] \mid \text{for all } [f_\ell, f^u] \in \mathcal{B}(\mathcal{F}_1) \text{ and } [g_\ell, g^u] \in \mathcal{B}(\mathcal{F}_2)\}$. First note for any $f_1 \in \mathcal{F}_1$ and $f_2 \in \mathcal{F}_2$, $f_1 \cap f_2$ is included in one of these brackets. In particular, for brackets $[f_\ell, f^u] \ni f_1$ and $[g_\ell, g^u] \ni f_2$, we have that $f_\ell \cap g_\ell \preceq f_1 \cap f_2 \preceq f^u \cap g^u$ and $[f_\ell \cap g_\ell, f^u \cap g^u] \in \mathcal{B}$. Furthermore,

$$\Pr_{x\sim\mu}\left[(f_\ell(x) \cap g_\ell(x)) \neq (f^u(x) \cap g^u(x))\right] \leq \Pr_{x\sim\mu}\left[(f_\ell(x) \cap g_\ell(x)) \neq (f_\ell(x) \cap g^u(x))\right]$$
$$+ \Pr_{x\sim\mu}\left[(f_\ell(x) \cap g^u(x)) \neq (f^u(x) \cap g^u(x))\right]$$
$$\leq 2\epsilon'.$$

Therefore, $\mathcal{B}$ is a $2\epsilon'$-bracketing for $\mathcal{F}_1 \cdot \mathcal{F}_2$ of size $\mathcal{N}_{[]}(\mathcal{F}_1, \mu, \epsilon') \cdot \mathcal{N}_{[]}(\mathcal{F}_2, \mu, \epsilon')$. Repeating this inductively and using $\epsilon' = \epsilon/k$, we get the claim for $k$ classes.

Similarly, for $\mathcal{F}_1 + \mathcal{F}_2$, construct $\mathcal{B} = \{[f_\ell \cup g_\ell, f^u \cup g^u] \mid \text{for all } [f_\ell, f^u] \in \mathcal{B}(\mathcal{F}_1) \text{ and } [g_\ell, g^u] \in \mathcal{B}(\mathcal{F}_2)\}$. First note for any $f_1 \in \mathcal{F}$ and $f_2 \in \mathcal{F}_1$ and their respective brackets $[f_\ell, f^u] \ni f_1$ and $[g_\ell, g^u] \ni f_2$, we have that $f_\ell \cup g_\ell \preceq f_1 \cup f_2 \preceq f^u \cup g^u$ and $[f_\ell \cup g_\ell, f^u \cup g^u] \in \mathcal{B}$. Furthermore,

$$\Pr_{x \sim \mu} \left[ (f_\ell(x) \cup g_\ell(x)) \neq (f^u(x) \cup g^u(x)) \right] \leq \Pr_{x \sim \mu} \left[ f_\ell(x) \neq f^u(x) \right] + \Pr_{x \sim \mu} \left[ g_\ell(x) \neq g^u(x) \right]$$

$$\leq 2\epsilon'.$$

Therefore, $\mathcal{B}$ is a $2\epsilon'$-bracketing for $\mathcal{F}_1 + \mathcal{F}_2$ of size $\mathcal{N}_{[]}(\mathcal{F}_1, \mu, \epsilon') \cdot \mathcal{N}_{[]}(\mathcal{F}_2, \mu, \epsilon')$. Repeating this inductively and using $\epsilon' = \epsilon/k$, we get the claim for $k$ classes.

As for the $\mathcal{G}$, the set of all symmetric differences, note that $f_1 \Delta f_2 = (f_1 \cup f_2) \setminus (f_1 \cap f_2) = (f_1 \cup f_2) \cap \overline{(f_1 \cap f_2)}$. Furthermore, for any class $\mathcal{F}$, the class $\overline{\mathcal{F}} = \{\overline{f} \mid \forall f \in \mathcal{F}\}$ has the same bracketing number as $\mathcal{F}$. Therefore, the bracketing number of $\mathcal{G}$ follows from using the bracketing number $\mathcal{F} + \mathcal{F}$, $\overline{\mathcal{F} + \mathcal{F}}$, and their intersection.

### C.6   Proof of Corollary 3.8

The set of polynomial threshold functions in $n$ variables and of degree $d$ is embeddable as halfspaces in $O(n^d)$ dimensions using the map

$$\phi(x_1, \ldots, x_n) = \left( \prod_{i \in S} x_i \right)_{S \in \{1, \ldots, n\}^{\leq d}},$$

which maps variables to all monomial of degree $d$. It can be seen that the number of monomials of degree at most $d$ in $n$ variables is given by $\binom{n+d+1}{d+1}$ which is approximately $O(n^d)$ when $d$ is small. Combining Theorem 3.6 and Theorem 3.4 completes the proof for polynomial threshold functions.

A $k$-polytope in $\mathbb{R}^n$ is an intersection of $k$-halfspaces in $\mathbb{R}^n$. Combining Theorem 3.7 and Theorem 3.4 completes the proof.

## D   More Details on Bracketing Number and Sign Rank

Though bracketing numbers are a fundamental concept in statistics, until recently their connection to VC theory was not well understood. Adams and Nobel [2010, 2012] show that for countable (can be generalized to classes that are well approximated by countable classes) classes with finite VC dimension the bracketing numbers with respect to any measure is finite (this establishes what is known as a universal Gilvenko–Cantelli theorem under ergodic sampling.)

**Theorem D.1** (Finite Bracketing Bounds for VC Classes)**.** *Let $\mathcal{C}$ be a countable class with finite VC dimension. Then, $\mathcal{N}_{[]}(\mathcal{C}, \mu, \epsilon) < \infty$.*

Though the above theorem proves that $\epsilon$-bracketing numbers are finite, their growth rate in $1/\epsilon$ can be arbitrarily large. See van Handel [2013] for some interesting examples of classes where the bracketing numbers grow arbitrarily fast.

Another combinatorial quantity that can help bound the regret in presence of adaptive smooth adversaries is *sign rank*.

**Definition D.2** (Sign Rank)**.** *Let $\mathcal{X}$ be an instance space and let $\mathcal{F}$ be a class. We can denote the class naturally as $\{-1, 1\}$-valued $\mathcal{X} \times \mathcal{F}$ matrix $M_\mathcal{F}$ where the entry corresponding to $(x, f)$ is $f(x)$. The sign rank of a class is the highest rank of a real matrix that agrees with a finite submatrix of $M_\mathcal{F}$ in sign. If this is unbounded, the class is said to have infinite sign rank.*

The sign rank of a class captures the dimension in which the class can be embedded as thresholds.

**Fact D.3** (Sign Rank Embedding, see e.g. Lokam [2009])**.** *The sign rank of a class corresponds to the smallest dimension $d$ that the class can be embedded as thresholds.*

Theorem 3.6 effectively says that classes with small sign rank have a slowly growing bracketing numbers and thus have low regret in the smoothed online learning setting. Thus, the complexity of smoothed online learning lies somewhere in between the sign rank and VC dimension. On the other

hand, it is known that even classes with small VC dimension can have arbitrarily large sign rank [Alon et al., 1987, Ben-David et al., 2003, Alon et al., 2016]. An intermediate question is whether classes with slow growing bracketing number also have good sign rank. It would be interesting to characterize the complexity of smoothed online learning in terms of either the sign rank or bracketing numbers.

# E  Query Answering

## E.1  Smooth MWEM Algorithm

---

**Algorithm 2:** Smooth Multiplicative Weights Exponential Mechanism

---

**Input:** Universe $\mathcal{X}$ with $|\mathcal{X}| = N$, Data set $B$ with $n$ records, Query set $\mathcal{Q}$, Privacy parameters $\epsilon$ and $\delta$, Smoothness parameter $\sigma$.

Let $\mathcal{D}_0(x) = 1/N$ for all $x \in \mathcal{X}$.
*Cover Construction:* Compute $\mathcal{Q}' \subseteq \mathcal{Q}$ that is a $\gamma$-cover of $\mathcal{Q}$ with respect to the uniform distribution for $\gamma = \frac{\sigma}{2n}$.
**for** $i = 1 \dots T$ **do**

    *Exponential Mechanism:* Sample $q_i \in \mathcal{Q}'$ according to the exponential mechanism with parameter $\epsilon/2T$ and score function

$$s_i(\mathcal{D}_B, q) = n \left| q\left(\mathcal{D}_{i-1}\right) - q(\mathcal{D}_B) \right|.$$

    *Laplace Mechanism:* Let $m_i = q_i(\mathcal{D}_B) + \frac{1}{n} Lap\left(2T/\epsilon\right)$.
    *Multiplicative Update:* Update $\mathcal{D}_{i-1}$ using the rule

$$\mathcal{D}_i(x) \propto \mathcal{D}_{i-1}(x) \exp\left( \frac{q_i(x)\left(m_i - q_i(\mathcal{D}_{i-1})\right)}{2} \right).$$

**end**
Let $\overline{\mathcal{D}} = \frac{1}{T} \sum_{i=1}^{T} \mathcal{D}_i$.
**Output:** For each $q \in \mathcal{Q}$, answer with $v_q = q'\left(\overline{\mathcal{D}}\right)$ where $q'$ is the closest function in $\mathcal{Q}'$ to $q$.

---

## E.2  Proof of Theorem 4.2

In this section we prove the following theorem.

**Theorem 4.2 (restated).** *For any $(\sigma, 0)$-smooth dataset $B$ of size $n$, a query class $\mathcal{Q}$ with VC dimension $d$, $T \in \mathbb{N}$ and $\epsilon > 0$, Smooth Multiplicative Weights Exponential Mechanism is $\epsilon$-differentially private and with probability at least $1 - 2T\left(\gamma/41\right)^{\mathrm{VCDim}(\mathcal{Q})}$, calculates values $v_q$ for all $q \in \mathcal{Q}$ such that*

$$\max_{q \in \mathcal{Q}} \{|v_q - q(\mathcal{D}_B)|\} \leq \frac{1}{n} + 2\sqrt{\frac{\log(1/\sigma)}{T}} + \frac{10Td\log(2n/\sigma)}{\epsilon n}.$$

Let us first provide a few useful lemmas.

**Lemma E.1** (Cover under Smoothness). *Let $B$ be $(\sigma, 0)$-smooth data set. Let $\mathcal{Q}' \subseteq \mathcal{Q}$ be a $\gamma$-cover of $\mathcal{Q}$ under the uniform distribution. For a $q \in \mathcal{Q}$, let $q' \in \mathcal{Q}$ be such that $\Pr_{x \sim \mathcal{U}}\left[q(x) \neq q'(x)\right] \leq \gamma$. Then,*

$$|q(\mathcal{D}_B) - q'(\mathcal{D}_B)| \leq \frac{2\gamma}{\sigma}.$$

*Proof.* From the $(\sigma, 0)$-smoothness of $B$, we get

$$|q(\mathcal{D}_B) - q'(\mathcal{D}_B)| = \left| q\left(\overline{\mathcal{D}_B}\right) - q'\left(\overline{\mathcal{D}_B}\right) \right|$$

$$\leq \sum_{x \in D} |(q(x) - q'(x))| \overline{\mathcal{D}_B}(x)$$

$$\leq \sum_{x \in \mathcal{X}} 2\mathbb{I}(q(x) \neq q'(x)) \overline{\mathcal{D}_B}(x)$$

$$\leq \frac{2}{\sigma} \sum_{x \in \mathcal{X}} \mathbb{I}(q(x) \neq q'(x)) \mathcal{U}(x)$$

$$\leq \frac{2}{\sigma} \Pr_{x \sim \mathcal{U}} [q(x) \neq q'(x)]$$

$$\leq \frac{2\gamma}{\sigma}$$

as required. $\qquad\qquad\square$

Define the potential function $\Psi_i = \sum_{x \in \mathcal{X}} \overline{\mathcal{D}_B}(x) \log \left( \overline{\mathcal{D}_B}(x) / \mathcal{D}_i(x) \right)$, where $\overline{\mathcal{D}_B}$ is a corresponding $\sigma$-smooth distribution that matches the query answers for the $(\sigma, 0)$-smooth data set $B$. Here we make a few observations about the potential function.

**Fact E.2.** *For all $i \leq T$, we have $\Psi_i \geq 0$. Furthermore, $\Psi_0 \leq \log \frac{1}{\sigma}$. As a result, $\Psi_0 - \Psi_T \leq \log \frac{1}{\sigma}$.*

*Proof.* The first claim follows from the positivity of the KL divergence. For the second one, recall that from the $\sigma$-smoothness of $\mathcal{D}_B$ and the fact that $\mathcal{D}_1$ is the uniform distribution, we have $\mathcal{D}_B(x) \leq \sigma^{-1} \mathcal{D}_0(x)$ for all $x \in \mathcal{X}$.

$$\Psi_0 = \sum_{x \in \mathcal{X}} \overline{\mathcal{D}_B}(x) \log \frac{\overline{\mathcal{D}_B}(x)}{\mathcal{D}_0(x)} \leq \sum_{x \in \mathcal{X}} \overline{\mathcal{D}_B}(x) \log \frac{1}{\sigma} = \log \frac{1}{\sigma}$$

as required. $\qquad\qquad\square$

Below is a direct adaptation of a result of Hardt et al. [2012] for bounding the change in the potential functions.

**Lemma E.3** (Lemma A.4 in Hardt et al. [2012]).

$$\Psi_{i-1} - \Psi_i \geq \left( \frac{q_i(\mathcal{D}_{i-1}) - q_i(\overline{\mathcal{D}_B})}{2} \right)^2 - \left( \frac{m_i - q_i(\overline{\mathcal{D}_B})}{2} \right)^2 .$$

**Lemma E.4** (Exponential and Laplace Mechanism guarantees). *With probability at least $1 - \frac{2T}{|\mathcal{Q}'|}$, we have*

$$|q_i(\mathcal{D}_{i-1}) - q_i(\mathcal{D}_B)| \geq \max_{q' \in \mathcal{Q}'} \{q'(\mathcal{D}_i) - q'(\mathcal{D}_B)\} - \frac{8T \log |\mathcal{Q}'|}{\epsilon n}$$

*and*

$$|m_i - q_i(\mathcal{D}_B)| \leq \frac{2T \log |\mathcal{Q}'|}{\epsilon n}.$$

Here we recall again the error guarantees from Hardt et al. [2012].

**Theorem E.5** (Hardt et al. [2012]). *For any data set $B$ of size $n$, a finite query class $\mathcal{Q}$, $T \in \mathbb{N}$ and $\epsilon > 0$, MWEM is $\epsilon$-differentially private and with probability at least $1 - \frac{2T}{|\mathcal{Q}|}$ produces a distribution $\overline{\mathcal{D}}$ over $\mathcal{X}$ such that*

$$\max_{q \in \mathcal{Q}} \left\{ |q(\overline{\mathcal{D}}) - q(\mathcal{D}_B)| \right\} \leq 2\sqrt{\frac{\log |\mathcal{X}|}{T}} + \frac{10T \log |\mathcal{Q}|}{\epsilon n}.$$

*Proof of Theorem 4.2.* Our proof closely resembles that of Theorem E.5 from Hardt et al. [2012]. Note that since $B$ is $(\sigma, 0)$-smooth, we have a $\sigma$-smooth distribution $\overline{\mathcal{D}_B}$ with $\overline{\mathcal{D}_B}(x) \leq \frac{1}{\sigma N}$ such that for all $q \in \mathcal{Q}$, $q(\mathcal{D}_B) = q(\overline{\mathcal{D}_B})$. Furthermore, note that we chose a cover $\mathcal{Q}' \subseteq \mathcal{Q}$. Therefore, $q'(\mathcal{D}_B) = q'(\overline{\mathcal{D}_B})$ holds for all $q' \in \mathcal{Q}'$ as well.

Note that since $q'(\mathcal{D}_B) = q'(\overline{\mathcal{D}_B})$ for all $q' \in \mathcal{Q}'$, we can replace this in the above equation. For the sake of completeness, we sketch the rest of the proof. From Jensen's inequality, we have

$$\max_{q' \in \mathcal{Q}'} \left| q'\left(\overline{\mathcal{D}}\right) - q'\left(\mathcal{D}_B\right) \right| \leq \frac{1}{T} \sum_{i=1}^{T} \max_{q' \in \mathcal{Q}'} \left| q'\left(\mathcal{D}_i\right) - q'\left(\mathcal{D}_B\right) \right|. \tag{7}$$

From Lemma E.4 and Lemma E.3, we get that with probability at least $1 - 2T/|\mathcal{Q}'|$, we get

$$\Psi_{i-1} - \Psi_i \geq \left( \frac{\max_{q' \in \mathcal{Q}'} \{ q'(\mathcal{D}_i) - q'(\mathcal{D}_B) \} - \frac{8T \log |\mathcal{Q}'|}{\epsilon n}}{2} \right)^2 - \left( \frac{T \log |\mathcal{Q}|}{\epsilon n} \right)^2.$$

Rearranging this and taking the average, we get

$$\frac{1}{T} \sum_{i=1}^{T} \max_{q' \in \mathcal{Q}'} \left| q'(\mathcal{D}_i) - q'(\mathcal{D}_B) \right| \leq \frac{1}{T} \sum_{i=1}^{T} \left[ \sqrt{4 \left( \Psi_{i-1} - \Psi_i \right) + \frac{4T^2 \log^2 |\mathcal{Q}'|}{n^2 \epsilon^2}} + \frac{8T \log |\mathcal{Q}'|}{n\epsilon} \right].$$

Applying the concavity of the square root function i.e., $\frac{1}{T} \sum_{i=1}^{T} (x_i)^{1/2} \leq \left( \frac{1}{T} \sum_{i=1}^{T} x_i \right)^{1/2}$,

$$\frac{1}{T} \sum_{i=1}^{T} \max_{q \in \mathcal{Q}'} \left| q'(\mathcal{D}_i) - q'(\mathcal{D}_B) \right| \leq \sqrt{\sum_{i=1}^{T} \frac{4 \left( \Psi_{i-1} - \Psi_i \right)}{T} + \frac{4T^2 \log^2 |\mathcal{Q}'|}{n^2 \epsilon^2}} + \frac{8T \log |\mathcal{Q}'|}{n\epsilon}$$

$$\leq \sqrt{\frac{4 \left( \Psi_0 - \Psi_T \right)}{T} + \frac{4T^2 \log^2 |\mathcal{Q}'|}{n^2 \epsilon^2}} + \frac{8T \log |\mathcal{Q}'|}{n\epsilon}$$

$$\leq \sqrt{\frac{4 \log \left( \frac{1}{\sigma} \right)}{T} + \frac{4T^2 \log^2 |\mathcal{Q}'|}{n^2 \epsilon^2}} + \frac{8T \log |\mathcal{Q}'|}{n\epsilon}$$

$$\leq 2\sqrt{\frac{\log \left( \frac{1}{\sigma} \right)}{T}} + \frac{10T \log |\mathcal{Q}'|}{n\epsilon}.$$

The second inequality follows by summing the telescoping series. The third follows from Fact E.2. The last equation follows from the fact that $\sqrt{x+y} \leq \sqrt{x} + \sqrt{y}$ for all positive $x, y$. Using Equation 7 and the fact that $|\mathcal{Q}|' \leq (41/\gamma)^d$ we have

$$\max_{q' \in \mathcal{Q}'} \left| q'\left(\overline{\mathcal{D}}\right) - q'\left(\mathcal{D}_B\right) \right| \leq 2\sqrt{\frac{\log (1/\sigma)}{T}} + \frac{10 T d \log (2n/\sigma)}{\epsilon n}.$$

Let $v_q = q'(\overline{\mathcal{D}})$ for $q' \in \mathcal{Q}'$ that is the closest hypothesis to $q$ with respect to the uniform distribution. Then

$$|q(\mathcal{D}_B) - v_q| = \left| q(\mathcal{D}_B) - q'(\mathcal{D}_B) + q'(\mathcal{D}_B) - q'\left(\overline{\mathcal{D}}\right) \right|$$

$$\leq \left| q(\mathcal{D}_B) - q'(\mathcal{D}_B) \right| + \left| q'(\mathcal{D}_B) - q'\left(\overline{\mathcal{D}}\right) \right|$$

$$\leq \frac{2\gamma}{\sigma} + 2\sqrt{\frac{\log 1/\sigma}{T}} + \frac{10 T d \log (41/\gamma)}{\epsilon n}.$$

Setting $\gamma = \frac{\sigma}{4n}$, we get the desired result. $\square$

Setting $T = \epsilon^{2/3} n^{2/3} \log^{1/3} (1/\sigma) d^{-2/3} \log^{-2/3} (2n/\sigma)$, we get $(\epsilon, 0)$ differential privacy with

$$\max_{q \in \mathcal{Q}} \{ |v_q - q(\mathcal{D}_B)| \} \leq O \left( \sqrt[3]{\frac{d \log (1/\sigma) \log (2n/\sigma)}{n\epsilon}} \right).$$

Also, as noted in Hardt et al. [2012], one can use adaptive $k$-fold composition (see e.g. Dwork and Roth [2014]) to get $(\epsilon, \delta)$-differential privacy with

$$\max_{q \in \mathcal{Q}} \{ |v_q - q(\mathcal{D}_B)| \} \leq O \left( \sqrt{\frac{d}{\epsilon n} \log^{\frac{1}{2}} \left( \frac{1}{\sigma} \right) \log \left( \frac{n}{\sigma} \right) \log \left( \frac{1}{\delta} \right)} \right).$$

### E.3 Running Time of the Algorithm

The running time of the algorithm is similar to the running time of the MWEM algorithm of Hardt et al. [2012]. The main additional step is the construction of the cover $\mathcal{Q}'$. Similar to Appendix C.1 , this cover can be constructed in time $O\left(|\mathcal{Q}'|\right)$. The exponential mechanism requires $O\left(n|\mathcal{Q}|'\right)$ to evaluate all the queries on the cover and time $O\left(|\mathcal{Q}|'|\mathcal{X}|\right)$ to execute each iteration of the algorithm. Recall that $|\mathcal{Q}'| \leq (41n/\sigma)^d$, thus the running time is bounded by $O\left(n\left(41n/\sigma\right)^d + T\left(41n/\sigma\right)^d|\mathcal{X}|\right)$.

This runtime can also be improved using several theoretical tricks, e.g., $q(\mathcal{D}_i)$ can be approximated by taking random points from $\mathcal{D}_i$ in time that is independent of $\mathcal{X}$.

Note that the runtime of our algorithm improves upon the runtime of MWEM by using smaller query sets. As noted in Hardt et al. [2012], their algorithm is amenable to many optimizations and modifications that make it very fast and practical Hardt et al. [2012].

## F  Data Release

### F.1  Projected Smooth MWEM Algorithm

---
**Algorithm 3:** Projected Smooth Multiplicative Weight Exponential Mechanism

---
**Input:** Universe $\mathcal{X}$ with $|\mathcal{X}| = N$, Data set $B$ with $n$ records, Query set $\mathcal{Q}$, Privacy parameters $\epsilon$ and $\delta$, Smoothness parameter $\sigma$.

Let $\mathcal{D}_0\left(x\right) = 1/N$ for all $x \in \mathcal{X}$.
*Cover Construction:* Compute $\mathcal{Q}' \subseteq \mathcal{Q}$ that is a $\gamma$-cover of $\mathcal{Q}$ with respect to the uniform distribution for $\gamma = \frac{\sigma}{2n}$.
**for** $i = 1 \ldots T$ **do**

   *Exponential Mechanism:* Sample $q_i \in \mathcal{Q}'$ according to the exponential mechanism with parameter $\epsilon/2T$ and score function

$$s_i(\mathcal{D}_B, q) = n\left|q\left(\mathcal{D}_{i-1}\right) - q(\mathcal{D}_B)\right|.$$

   *Laplace Mechanism:* Let $m_i = q_i\left(\mathcal{D}_B\right) + \frac{1}{n}Lap\left(2T/\epsilon\right)$.
   *Multiplicative Update:* Update $\mathcal{D}_{i-1}$ using the rule

$$\tilde{\mathcal{D}}_i\left(x\right) \propto \mathcal{D}_{i-1}\left(x\right)\exp\left(\frac{q_i\left(x\right)\left(m_i - q_i(\mathcal{D}_{i-1})\right)}{2}\right).$$

   *KL Projection:* Project $\tilde{\mathcal{D}}_i$ onto the polytope $\mathcal{K} = \left\{\mathbf{z} : z_i \geq 0,\ \sum_{i=1}^{N} z_i = 1, z_i \leq \frac{1}{\sigma N}\right\}$ of

   smooth distributions:
$$\mathcal{D}_i = \underset{\mathcal{D}\in\mathcal{K}}{\arg\min}\,\mathrm{D}_{\mathrm{KL}}(\mathcal{D}\|\tilde{\mathcal{D}}_i)$$

**end**
Let $\overline{\mathcal{D}} = \frac{1}{T}\sum_{i=1}^{T}\mathcal{D}_i$.
**Output:** Distribution $\overline{\mathcal{D}}$.

---

### F.2  Proof of Theorem 4.3

As before, let $\overline{\mathcal{D}_B}$ be a corresponding $\sigma$-smooth distribution that matches the query answers for the $(\sigma, 0)$-smooth data set $B$. Define $\Psi_i = \sum_{x\in\mathcal{X}}\overline{\mathcal{D}_B}(x)\log\left(\overline{\mathcal{D}_B}(x)/\mathcal{D}_i(x)\right)$ and $\tilde{\Psi}_i = \sum_{x\in\mathcal{X}}\overline{\mathcal{D}_B}(x)\log\left(\overline{\mathcal{D}_B}(x)/\tilde{\mathcal{D}}_i(x)\right)$ as the intermediate potential. From Lemma E.3, we know

$$\Psi_{i-1} - \tilde{\Psi}_i \geq \left(\frac{q_i\left(\mathcal{D}_{i-1}\right) - q_i(\mathcal{D}_B)}{2}\right)^2 - \left(\frac{m_i - q_i(\mathcal{D}_B)}{2}\right)^2.$$

Using the properties of relative entropy, we show the following claim.

**Claim F.1.** *For every $i \leq T$, we have $\tilde{\Psi}_i \geq \Psi_i$.*

*Proof.* The claim follows from the following fact about the KL divergence. Let

$$\mathcal{D}_i = \operatorname*{argmin}_{\mathcal{D} \in \mathcal{K}} \mathrm{D}_{\mathrm{KL}}(\mathcal{D} \| \tilde{\mathcal{D}}_i)$$

for some convex set $\mathcal{K}$. Then, for $\overline{\mathcal{D}_B} \in \mathcal{K}$,

$$\mathrm{D}_{\mathrm{KL}}(\overline{\mathcal{D}_B} \| \tilde{\mathcal{D}}_i) \geq \mathrm{D}_{\mathrm{KL}}\left(\overline{\mathcal{D}_B} \| \mathcal{D}_i\right) + \mathrm{D}_{\mathrm{KL}}\left(\mathcal{D}_i \| \tilde{\mathcal{D}}_i\right).$$

The claim follows by $\tilde{\Psi}_i = \mathrm{D}_{\mathrm{KL}}(\mathcal{D}_B \| \tilde{\mathcal{D}}_i)$, $\Psi_i = \mathrm{D}_{\mathrm{KL}}\left(\mathcal{D}_B \| \mathcal{D}_i\right)$ and $\mathrm{D}_{\mathrm{KL}}\left(\mathcal{D}_i \| \tilde{\mathcal{D}}_i\right) \geq 0$. $\square$

Together this gives

$$\Psi_{i-1} - \Psi_i \geq \left(\frac{q_i\left(\mathcal{D}_{i-1}\right) - q_i(\mathcal{D}_B)}{2}\right)^2 - \left(\frac{m_i - q_i(\mathcal{D}_B)}{2}\right)^2.$$

The remainder of the analysis follows that of Theorem 4.2. Note that we have $\overline{\mathcal{D}}$ is $\sigma$-smooth since each $\mathcal{D}_i \in \mathcal{K}$ and $\mathcal{K}$ is a convex set. By Lemma E.1, we have $\left|q'\left(\overline{\mathcal{D}}\right) - q\left(\overline{\mathcal{D}}\right)\right| \leq {}^{2\gamma}/\sigma$. Thus,

$$
\begin{aligned}
\left|q\left(\mathcal{D}_B\right) - q\left(\overline{\mathcal{D}}\right)\right| &= \left|q\left(\mathcal{D}_B\right) - q'\left(\mathcal{D}_B\right) + q'\left(\mathcal{D}_B\right) - q'\left(\overline{\mathcal{D}}\right) + q'\left(\overline{\mathcal{D}}\right) - q\left(\overline{\mathcal{D}}\right)\right| \\
&\leq \left|q\left(\mathcal{D}_B\right) - q'\left(\mathcal{D}_B\right)\right| + \left|q'\left(\mathcal{D}_B\right) - q'\left(\overline{\mathcal{D}}\right)\right| + \left|q'\left(\overline{\mathcal{D}}\right) - q\left(\overline{\mathcal{D}}\right)\right| \\
&\leq \frac{4\gamma}{\sigma} + 2\sqrt{\frac{\log {}^1/\sigma}{T}} + \frac{10Td\log\left({}^{41}/\gamma\right)}{\epsilon n}.
\end{aligned}
$$

Setting $\gamma = {}^{\sigma}/{4n}$, we get

$$\left|q\left(\mathcal{D}_B\right) - q\left(\overline{\mathcal{D}}\right)\right| = \frac{1}{n} + 2\sqrt{\frac{\log\left({}^1/\sigma\right)}{T}} + \frac{10Td\log\left({}^{4n}/\sigma\right)}{\epsilon n}.$$

### F.3 Running Time of Projected Smooth Multiplicative Weights Exponential Mechanism

The running time is similar to the running time Smooth Multiplicative Weights Exponential Mechanism, with the additional projection step in each step. Note that the projection in each step is a convex program and can be solved in time $\mathrm{poly}\left(|\mathcal{X}|\right)$. This gives us a total running time of $O\left(n\left(41n/\sigma\right)^d + T\left(41n/\sigma\right)^d |\mathcal{X}| + T\mathrm{poly}(|\mathcal{X}|)\right)$.

In addition to the improvements discussed in the previous sections, the projection step can be performed faster by taking an approximate Bregman projection as considered by Barak et al. [2009]. Incorporating this into our algorithm would lead to significant speed ups.

# G Smooth Data Release using SmallDB Algorithm

In this section,, we look at a different algorithm to get differential privacy when dealing with $(\sigma, \chi)$-smooth data sets. Our algorithm displayed below uses several pieces that have been introduced by Blum et al. [2008] and Hardt and Rothblum [2010].

---

**Algorithm 4:** Subsampled Net Mechanism

---

**Input:** Database $B$ of size $n$, Query set $\mathcal{Q}$, Privacy parameter $\epsilon$, Subsampling parameter $M$, Accuracy parameter $\gamma$.

Sample (with replacement) a subset $V$ of size $M$ from $\mathcal{X}$.

Sample $B'$ from amongst all data sets supported on $V$ of size

$$O\left(\frac{d}{\gamma^2}\right)$$

with probability proportional to

$$\exp\left(-\frac{\epsilon \cdot n \cdot s\left(\mathcal{D}_{B'}, \mathcal{D}_B\right)}{2}\right)$$

where $s\left(\mathcal{D}_{B'}, \mathcal{D}_B\right) = \max_{q \in \mathcal{Q}} |q(\mathcal{D}_B) - q(\mathcal{D}_{B'})|$.

**Output:** Database $B'$

---

First, we analyze the privacy of this algorithm.

**Theorem G.1.** *The Subsampled Net Mechanism is $(\epsilon, 0)$ differentially private.*

*Proof.* The privacy claim follows from the privacy of the exponential mechanism. $\square$

Next we bound the error of this mechanism. Let us recall the standard uniform convergence bound.

**Fact G.2** (Uniform Convergence for VC Classes, see e.g. Shalev-Shwartz and Ben-David [2014])**.** *Let $\mathcal{X}$ be the domain, $\mathcal{Q}$ be a class of queries over $\mathcal{X}$ with VC dimension $d$ and let $\mathcal{D}$ be a distribution. Let $\mathcal{D}'$ be a distribution gotten by sampling $O\left((\log(2/\eta) + d)/\gamma^2\right)$ items iid from $\mathcal{D}$ and normalizing the frequencies. Then, with probability $1 - \eta$, for all $q \in \mathcal{Q}$, $|q(\mathcal{D}') - q(\mathcal{D})| \leq \gamma$.*

In the following, we use the above fact to show that a randomly sampled subset of the universe approximates a $(\sigma, \chi)$-smooth database. The proof largely follows the domain reduction lemma of Hardt and Rothblum [2010] that achieve a similar bond by with a dependence on $\log(|\mathcal{Q}|)$. We include this proof for completeness.

**Lemma G.3.** *Let $\mathcal{X}$ be a data universe and $\mathcal{Q}$ a collection of queries over $\mathcal{X}$ with VC dimension $d$ and $\mathcal{D}$ be $(\sigma, \chi)$-smooth with respect to $\mathcal{Q}$. Let $V \subset \mathcal{X}$ of size $M$ be sampled from $\mathcal{X}$ at random with replacement with*

$$M = O\left(\frac{\log(1/\eta) + d}{\sigma \gamma^2}\right).$$

*Then, with probability $1 - \eta$, there exists a $\mathcal{D}'$ on $V$ such that for all $q \in \mathcal{Q}$*

$$|q(\mathcal{D}) - q(\mathcal{D}')| \leq \chi + \gamma.$$

*Proof.* Let $\mathcal{D}_1$ be $\sigma$-smooth distribution that witnesses the $(\sigma, \chi)$-smoothness of $\mathcal{D}$. If we could sample from $\mathcal{D}_1$, we would be done from Fact G.2. But we want to get a subset that is oblivious to the distribution $\mathcal{D}$. To achieve this, we use the smoothness of $\mathcal{D}_1$.

The idea is to sample from $\mathcal{D}_1$ using rejection sampling. Since $\mathcal{D}_1$ is $\sigma$-smooth, the following procedure produces samples from $\mathcal{D}_1$: sample from the uniform distribution and accept sample $u$ with probability $\sigma N \mathcal{D}_1(u)$. Note that accepted samples are distributed according to $\mathcal{D}_1$. We repeat this process until $O\left((\log(2/\eta) + d)/\gamma^2\right)$ samples are accepted. Since the accepted samples are distributed according to $\mathcal{D}_1$, from Fact G.2, there is a distribution $\mathcal{D}_2$ supported on the accepted samples such that with probability at least $1 - \eta/2$ for all $q \in \mathcal{Q}$,

$$|q(\mathcal{D}_2) - q(\mathcal{D})| \leq \chi + \gamma.$$

Let $S_1$ be the coordinates corresponding to the accepted samples and $S_2$ be the coordinates corresponding to the rejected ones. The key observation is that $S = S_1 \cup S_2$ is subset generated by sampling from the uniform distribution and has a distribution supported on it that approximates $\mathcal{D}$. So, it suffices to bound the size of $S$. The probability that a given sample gets accepted is

$$\sum_{x \in \mathcal{X}} \frac{\mathcal{D}_1(x) N\sigma}{N} = \sigma.$$

Thus the expected number of samples needed in the rejection sampling procedure is $M = O\left(\frac{\log(2/\eta)+d}{\sigma\gamma^2}\right)$. Using a Chernoff bound, we can bound the probability that this is greater than its mean by a factor of $4$ by

$$e^{-M} \leq \frac{\eta}{2}$$

where we used that fact that $M \geq \log(2/\eta)$. $\qquad\square$

We are finally ready to prove our theorem.

**Theorem G.4.** *For any data set $B$ that is $(\sigma, \chi)$-smooth with respect to a set of queries $\mathcal{Q}$ of VC dimension $d$, the output $\mathcal{D}''$ of the Subsampled Net Mechanism satisfies that with probability $1 - \eta$, for all $q \in \mathcal{Q}$*

$$|q(\mathcal{D}_B) - q(\mathcal{D}'')| \leq \chi + \tilde{O}\left(\sqrt[3]{\frac{d\log(1/\sigma) + \log(1/\eta)}{\epsilon n}}\right)$$

*Proof.* Consider a subset $V$ sampled with size $M = O\left(\frac{\log(1/\eta_1)+d}{\sigma\gamma^2}\right)$ where $\eta_1$ and $\gamma$ are parameters we will set later. From Lemma G.3, with probability $1 - \eta_1$ we have that there exists a distribution $\mathcal{D}'$ supported on $V$ such that for all $q \in \mathcal{Q}$

$$|q(\mathcal{D}') - q(\mathcal{D}_B)| \leq \chi + \gamma.$$

Let us work conditioned on this event. Let $A$ denote the set of all data sets supported on $V$ and let $C$ denote all data sets supported on $V$ with size $O\left(d\gamma^{-2}\right)$. From Fact G.2, for any distribution $\mathcal{D}_1$ supported on $V$, there is a data set in $C$ whose distribution $\mathcal{D}_2$ satisfies

$$|q(\mathcal{D}_1) - q(\mathcal{D}_2)| \leq \gamma.$$

We recall the guarantees of the exponential mechanism (see e.g. Dwork and Roth [2014]): Let $B''$ be the data base output by the exponential mechanism. Then,

$$\Pr\left[s(\mathcal{D}_{B''}, \mathcal{D}_B) \geq \min_{B_1 \in C} s(\mathcal{D}_{B_1}, \mathcal{D}_B) - \frac{2}{\epsilon n}(\log|C| + t)\right] \leq e^{-t},$$

where $s(\mathcal{D}_B, \mathcal{D}_{B'}) = \max_{q \in \mathcal{Q}}|q(\mathcal{D}_B) - q(\mathcal{D}_{B'})|$. Note that $\log|C| \leq M^{O(d\gamma^{-2})}$. Thus, with probability $1 - \eta_2$,

$$s(\mathcal{D}_{B''}, \mathcal{D}_B) \geq \min_{B_1 \in C} s(\mathcal{D}_{B_1}, \mathcal{D}_B) - \gamma$$

for

$$\gamma \geq \frac{4}{\epsilon n} \log \frac{M^{O(d\gamma^{-2})}}{\eta_2}.$$

Since, $\min_{B_1 \in C} s(\mathcal{D}_{B_1}, \mathcal{D}_B) \leq \chi + 2\gamma$, setting $\eta_1 = \eta_2 = \eta/2$ and solving for $\gamma$, we get

$$\gamma = \tilde{O}\left(\sqrt[3]{\frac{d\log(1/\sigma) + \log(1/\eta)}{\epsilon n}}\right)$$

as required. $\qquad\square$

## G.1 Running Time of Subsampled Net Mechanism

The running time of the algorithm involves first sampling $M$ elements uniformly from the domain which takes time $O(M\log|\mathcal{X}|)$. Each query needs to be evaluated on the data set $B$ which takes time $n|\mathcal{Q}|$. Evaluating and sampling from all data bases as required by the exponential mechanism naively takes time $M^{O(d\gamma^{-2})}$. As discussed earlier, this can be sped up using sampling for approximation.

## Footnotes

[3]Alternatively, we can bound $|\mathcal{H}'| \leq (41/\epsilon)^{\text{VCDim}(\mathcal{H})}$ by Haussler [1995].