[Reviews · NeurIPS 2020]

Review 1

Summary and Contributions: This paper revisits the problem of online classification and differentially private learning and data release. Both of these problems are typically studied in worst-case data models, in which the data is chosen by an adversary. It turns out that both problems are non-trivially solvable in this adversarial model if and only if the class of functions to be learned/summarized has finite Littlestone dimension. This is -bad news- over infinite data domains, because Littlestone dimension can be unbounded even when VC-dimension is constant, as in the case of threshold functions over a real interval. This paper makes an important conceptual contribution: it shows that in a hybrid adversarial/stochastic model, in which the adversary is restricted to choosing distributions whose density function is upper bounded by some multiple of the density under the uniform distribution, these problems can again be solved with sample/round complexity that depends on the VC-dimension of the class. There are a couple of nuances; the VC-dimension bounds only hold in the online learning setting against a non-adaptive adversary. Against an adaptive adversary, the authors are only able to prove bounds depending on the bracketing number of the concept class, but they show that for a natural family of functions (polynomial threshold functions), the bracketing number is bounded by the VC-dimension. Whether or not similar bounds can be proven against adaptive adversaries with dependence only on VC-dimension is an interesting question left open.

Strengths: The main strength of the paper is its conceptual message: that private and online learning are actually not as hard as one might have thought under mild assumptions. The proof of the results is relatively straightforward once one thinks to prove them (but they are not without subtlety, and in particular, handling the case of an adaptive adversary in an online learning setting is non-trivial/non-obvious). But I think this is a case of "clear only in hindsight" which is my favorite kind of result.

Weaknesses: The results for adaptive adversaries are less clean; its not clear whether bracketing number is the "right" measure or if the VC-dimension results for non-adaptive adversaries actually also hold for adaptive adversaries.

Correctness: Yes --- the claims are intuitive and clear.

Clarity: Yes

Relation to Prior Work: Yes

Reproducibility: Yes

Additional Feedback:


Review 2

Summary and Contributions: In this paper, the authors provide theoretical guarantees for the online and differentially private learning of a class of hypotheses H from samples (x, y), where x is an instance and y is a label. It has been proven recently that a hypotheses class H is online or differentially private PAC learnable if and only if it has finite Littlestone dimension. This is mainly an impossibility result since even linear threshold functions have infinite Littlestone dimension. The main result of the paper is that if we make additional assumptions on the distribution D of the instances x, then we can bypass the impossibility of Littlestone dimension. Namely the authors prove that the number of samples that we need to online or differentially private learn a class H depends in this case in the VC dimension of H which is the same as the offline and non-private learning of H. The assumption that the authors make on the distribution D of the instances x is that the probability density function of the distribution D is upper bounded by 1/sigma times the density of the uniform distribution.

Strengths: - The fact that the Littlestone dimension can be bypassed if we drop the assumption of worst-case distribution over the instances x is a very important contribution and it seems to have a lot of important applications in the future. - Technically the idea of using the notion of bracketing number is novel and of independent interest.

Weaknesses: 1. The assumption that the density is upper bounded by 1/sigma times the uniform is a relatively strong assumption when the instances x are high- dimensional. Imagine that the space of instances X is [0, 2]^d then the uniform distribution has density 1/2^d and hence the probability density over X is assumed to have density in the range [0, (1/sigma) (1/2^d)], which is a strong restriction for the distributions over X. 2. I don't think that the distributional assumption resembles the smoothed analysis model of Spielman and Teng. In the present paper the assumption of the authors is that the distribution over X is very close to the completely uniform distribution. In other words we start with the uniform distribution over X and then an adversary can move a small amount of mass in a worst-case way. On the other hand in Spielman and Teng we start from a worst-case instance and then we add a small amount of noise around that particular worst-case instance. An assumption that feels closer to the Spielman and Teng model would be that we start with an arbitrary distribution D and then we take the convolution of D with a Gaussian distribution or a uniform distribution over a ball with small radius. Post-Rebuttal ============================ Thanks a lot for the clarifications! I think a discussion of these points in the paper would help a lot. I change my score to strong accept.

Correctness: The main theoretical results of the paper seem correct.

Clarity: The paper is mostly well written although a more careful discussion on the assumptions that the authors use would be helpful.

Relation to Prior Work: yes

Reproducibility: Yes

Additional Feedback: I believe that the contribution of this paper is very important both because of the result but also because of the techniques that the authors use. I strongly recommend acceptance.


Review 3

Summary and Contributions: This paper studies the very interesting question of moving beyond worst-case adversarial bounds for private/online learning. This is of particular relevance since the question of the equivalence between private <-> online learning was recently resolved by Bun et al. 20, and this is a logical next step in this line of research. Rather than consider a worst case adversaries, the authors consider adversaries constrained to play instances from alpha-smooth distributions (hence smoothed analysis). Although smooth adversaries against online or private learning have been studied in specific instances, this is the first work to consider this problem in full generality. Results: -They show that online learning against smooth adversaries can be characterized by the bracketing number of the hypothesis class - And that private learning against smoothed adversaries can be characterized by the VC dimension of the hypothesis class. In the non-adaptive case, a no-regret algorithm against smoothed adversaries is known from Hagtalab 2018, which relies on a uniform convergence result wrt to regret over the hypothesis space, which doesn't hold in the adaptive setting. The key insight that extends this result to hold against adaptive adversaries, is that instead of taking an epsilon cover with respect to the uniform distribution, we take a cover consisting of the upper functions in an epsilon bracketing of the hypothesis space, which leads to the notion of bracketing number govering the regret bound. Similar techniques extend differentially private query answering, which suffers from lower bounds that include log terms on the size of the query class and data universe, to the infinite query class / data set size case. This makes sense because classical algorithms for solving these problems like MWEM are already cast in an online framework.

Strengths: This paper is well-motivated, and solves several problems of substantial interest to the core differential privacy / learning theory communities. In particular, the first comprehensive smoothed analysis of online learning against adaptive adversaries is a major result, of substantial interest to Neurips. The proofs are relatively easy to understand and verify. The use of bracketing number to characterize complexity, and the key lemma 3.2 seem like useful techniques.

Weaknesses: - The paper would be stronger if they provided matching lower bounds, showing that the bracketing number for online learning and vc dim for private learning were the right quantities to characterize learnability against smoothed adversaries. Nevertheless, since they show many common classes have finite bracketing number, their upper bounds clearly have utility greater than what was previously known. - The paper would be stronger with some empirical results / simulation studies evaluating the proposed algorithms against smooth distributions - The section on differentially private data / query release could be a little longer / more clearly described.

Correctness: Yes

Clarity: Yes

Relation to Prior Work: Yes

Reproducibility: Yes

Additional Feedback:

[Author Response · NeurIPS 2020]

We thank all the reviewers for their thoughtful feedback. We are happy to see that they agree with us on the importance of understanding online learning and differential privacy from the perspective of smoothed analysis. We will incorporate their feedback in the final version of the paper.

Below, we address the questions of **Reviewer #3** regarding the definition of smoothness.

Regarding the smoothness assumption (1), while we assume that $1/\sigma$ is a constant in $T$, we can use values of $\sigma$ that depend on the dimension of the space to account for the volume of high dimensional domains [1]. Since our regret bounds are only logarithmic in $1/\sigma$, they gracefully scale with the dimension of the space (which in the case of halfspaces, polynomial thresholds, etc, is bounded as a function of the VC dimension). In the context of the example mentioned in the review, note that $\sigma = c2^{-d}$ is only a weak smoothness assumption and still leads to a regret bound of $\tilde{O}\left(\sqrt{Td \cdot \text{VCDim}(\mathcal{F})}\right)$ against non-adaptive adversaries. Similar bounds hold against adaptive adversaries when the $\epsilon$-bracketing number grows comparably to the size of a classical $\epsilon$-cover, which as we argued in the paper is the case for many classes including halfspaces, polynomial thresholds, and convex polytopes.

Next we address (2) and compare our definition to Speilman and Teng's definition of smoothness.

Let us start by highlighting the similarities between our model of smoothness and Speilman and Teng's. Both of these models upper bound the maximum density within a domain. Our smoothness assumption does this directly by bounding the density by $1/\sigma$ times that of the uniform distribution. On the other hand, Speilman and Teng's model does this indirectly. To illustrate this, let $\mathcal{X}$ be the unit ball in $\mathbb{R}^d$ and take a Gaussian convolution with variance $O(1/d)$. For ease of presentation, clip this Gaussian convolution at a ball of radius 2. [2] It is not hard to see that the maximum density achieved anywhere in this ball is at most approximately $O(d)^{d/2}$, which is within an $\exp(d)$ factor of the uniform density on the ball of radius 2 (or even the unit ball) as required by our model of smoothness. Again, because of the logarithmic dependence on the value of $1/\sigma$, this leads to a regret bound of $\tilde{O}(\sqrt{Td \cdot \text{VCDim}(\mathcal{F})})$, against a non-adaptive smooth adversary. Note that, we have not optimized these parameters and indeed better translation between the two models of smoothness is possible when you choose the clipping region more carefully. This shows that the smoothness framework of Speilman and Teng is captured by the smoothness framework we use in this paper (within reasonable parameters).

We believe there are additional advantages in using our model of smoothness when formalizing learnability in a general sense. One advantage is that our smoothing model treats combinatorial domains (say $[n]$ or graphs on $n$ vertices) and geometric domains (say $D \subseteq \mathbb{R}^d$) in a unified manner and allows us to deliver results most meaningful to the analysis of learnability in presence of some smoothness without getting bogged down with domain-specific definitions of smoothness. Another advantage of our model — which is specifically important in machine learning — is that it naturally allows the adversary to have arbitrary correlations between the labels and the instances as long as the marginal on the instances is a "smooth" distribution. This can be handled by other models, albeit with more awkwardness in separating how an instance is generated by random shifts but its label is generated exactly by the adversary.

We agree that these are important discussion points and we will incorporate a summary of them in the final version of the paper.

## Footnotes

[1] Our results also extend to values of $1/\sigma$ that are subexponential in $T$

[2] Variance of $1/d$ is chosen so that the noise is of the same order as size of the domain, i.e., Gaussian distribution with variance $1/d$ is close to a uniform distribution over the unit ball. This clipping is done for ease of presentation and to ensure that no matter what the original distribution over $\mathcal{X}$ was, the Gaussian convolution has most of its density within the ball of radius 2. Qualitatively similar results can be achieved even without this clipping or with clipping to $\mathcal{X}$ which would be necessary if we restrict the adversary to only use instances in $\mathcal{X}$.


[Meta-Review · NeurIPS 2020]

Circumvening hardness through a smoothed analysis, the paper provides improved regret bounds and privacy guarantees for data release in this slightly easier model. The reviewers found the paper well written and the results interesting and of potential impact.